# T-cell co-stimulation in combination with targeting FAK drives enhanced anti-tumor immunity

**Marta Canel[1], David Taggart[1], Andrew H Sims[2], David W Lonergan[1], Irene C Waizenegger[3], Alan Serrels[1]***

[1]Centre for Inflammation Research, Queen's Medical Research Institute, University of Edinburgh, Edinburgh, United Kingdom; [2]Cancer Research UK Edinburgh Centre, MRC Institute of Genetics & Molecular Medicine, University of Edinburgh, Edinburgh, United Kingdom; [3]Boehringer Ingelheim RCV GmbH & Co KG, Vienna, Austria

**Abstract** Focal Adhesion Kinase (FAK) inhibitors are currently undergoing clinical testing in combination with anti-PD-1 immune checkpoint inhibitors. However, which patients are most likely to benefit from FAK inhibitors, and what the optimal FAK/immunotherapy combinations are, is currently unknown. We identify that cancer cell expression of the T-cell co-stimulatory ligand CD80 sensitizes murine tumors to a FAK inhibitor and show that CD80 is expressed by human cancer cells originating from both solid epithelial cancers and some hematological malignancies in which FAK inhibitors have not been tested clinically. In the absence of CD80, we identify that targeting alternative T-cell co-stimulatory receptors, in particular OX-40 and 4-1BB in combination with FAK, can drive enhanced anti-tumor immunity and even complete regression of murine tumors. Our findings provide rationale supporting the clinical development of FAK inhibitors in combination with patient selection based on cancer cell CD80 expression, and alternatively with therapies targeting T-cell co-stimulatory pathways.

**\*For correspondence:**
a.serrels@ed.ac.uk

## Introduction

Focal Adhesion Kinase (FAK) is a non-receptor protein tyrosine kinase that plays a pleotropic role in regulating cancer development and progression. Initially identified as a protein highly phosphorylated in response to integrin activation and primarily located at cell–extracellular matrix adhesion sites termed focal adhesions, FAK is now known to regulate a number of cellular processes linked to the malignant phenotype including migration, invasion, adhesion, polarity, proliferation, and survival (*McLean et al., 2005*; *Sulzmaier et al., 2014*). Furthermore, FAK can also translocate to the nucleus of cancer cells where it regulates inflammatory gene expression programs (*Lim et al., 2012*). This includes chemokines and cytokines, which orchestrate the composition of tumor microenvironment (TME), promoting immune evasion and resistance to immunotherapy at least in some scenarios (*Jiang et al., 2016*; *Serrels et al., 2015*; *Serrels et al., 2017*). Thus, FAK has emerged as a potentially promising target as a modulator of anti-cancer immunotherapy.

A number of small molecule FAK kinase inhibitors have now either completed, or are currently in, Phase-I clinical trials and results thus far suggest that they are generally well tolerated (*de Jonge et al., 2019*; *Doi et al., 2019*; *Lee et al., 2015*; *Soria et al., 2012*). Using panels of cancer cell lines, several studies have attempted to identify biomarkers that predict sensitivity to FAK inhibition. Loss of expression of the tumor suppressor protein Merlin (also known as Neurofibromin 2) and/or E-cadherin have both been suggested as potential markers of sensitivity to FAK inhibitors (*Hirt et al., 2018*; *Kato et al., 2017*; *Shah et al., 2014*; *Shapiro et al., 2014*). However, a clinical trial

(ClinicalTrials.gov NCT01870609) testing Merlin as a predictive marker for sensitivity to the FAK inhibitor Defactinib did not prove successful. Therefore, biomarkers enabling identification of patient populations most likely to benefit from FAK kinase inhibitors are much needed.

The discovery of a role for FAK in regulating the immuno-suppressive TME in mouse models of skin squamous cell carcinoma (SCC) (*Serrels et al., 2015*) and pancreatic cancer (*Jiang et al., 2016*; *Stokes et al., 2011*) has resulted in a shift in the clinical development of FAK kinase inhibitors, with new focus towards immune oncology. There are now at least two clinical trials testing FAK kinase inhibitors in combination with immune checkpoint inhibitors targeting anti-PD-1, across multiple tumor types including pancreatic cancer, non-small cell lung cancer and mesothelioma (ClinicalTrials.gov NCT02758587, NCT02546531). These trials stem from our previous work showing that FAK knockout or the FAK kinase inhibitor VS-4718 can drive complete CD8 T-cell dependent regression of established SCC tumors through shifting the balance of CD8 T-cells: Regulatory T-cells (Tregs) in favor of tumor clearance (*Serrels et al., 2015*), and that of others showing that VS-4718 can drive reprogramming of the immuno-suppressive pancreatic TME, including a reduction in Tregs, CD206$^+$ macrophages, myeloid-derived suppressor cells (MDSCs), and stromal fibroblasts. This renders pancreatic tumors responsive to the combination of dual immune checkpoint therapy and chemotherapy (anti-PD-1, anti-CTLA-4, and Gemcitabine) (*Jiang et al., 2016*).

Here, we show that expression of the T-cell co-stimulatory ligand CD80 on the surface of murine cancer cells sensitizes tumors to the highly selective and potent FAK kinase inhibitor BI 853520 (*Hirt et al., 2018*). Extrapolating these findings to human cancer, we show that murine CD80$^+$ SCC cells that are highly responsive to BI 853520 co-express a number of genes associated with a cancer stem phenotype that has previously been identified in human SCC tumors (*Miao et al., 2019*; *Oshimori et al., 2015*) and that a substantial proportion of human cancer cell lines representing a broad range of cancer types express the *CD80* transcript, supporting the potential for patient stratification based on cancer cell CD80 expression. Using murine CD80 negative SCC and pancreatic cancer cell lines that exhibit little response to BI 853520, we show that the combination of BI 853520 together with agonistic antibodies targeting other T-cell co-stimulatory receptors, in particular OX40 and 4-1BB, results in enhanced anti-tumor immunity and even complete CD8 T-cell dependent tumor regression leading to lasting immunological memory. Contributing to the enhanced anti-tumor efficacy of these combinations, we identify a novel role for FAK in regulating the expression of the immune checkpoint ligand PD-L2 on tumor-associated macrophages, monocytic-myeloid-derived suppressor cells (M-MDSCs) and cancer cells, and in regulating expression of the immune co-stimulatory receptor Inducible T-cell costimulator (ICOS) on effector CD8 T-cells. Therefore, FAK inhibition promotes greater responsiveness to the anti-tumor effects of T-cell co-stimulation through reprogramming multiple immune regulatory pathways, supporting further development of these combinations for clinical testing.

## Results

### Spectrum of responses to BI 853520

We have previously shown using a murine model of skin SCC that depletion of FAK expression or treatment with a small molecule FAK kinase inhibitor can result in immune-mediated tumor regression in syngeneic mice (*Serrels et al., 2015*). Using this same model system we first determined the anti-tumor efficacy of a different FAK kinase inhibitor, that is BI 853520 (*Hirt et al., 2018*), by monitoring tumor growth following injection of FAK-deficient cells (FAK-/-) or FAK-deficient cells that re-expressed wild-type FAK (FAK-wt) at comparable levels to endogenous. Daily treatment of SCC FAK-wt tumors with 50 mg/kg BI 853520 resulted in complete tumor regression with similar kinetics to that of SCC FAK-/-tumors (*Figure 1A*). Treatment of SCC FAK-/-tumors with BI 853520 had no effect on tumor growth.

Having established that treatment of SCC FAK-wt tumors with BI 853520 could recapitulate our previously published observations with a different FAK inhibitor (*Serrels et al., 2015*), we next set out to further investigate the generality of such therapeutic efficacy using a panel of six syngeneic cancer cell lines derived from three commonly used mouse cancer models: (1) skin squamous cell carcinomas induced using the DMBA/TPA two-stage chemical carcinogenesis protocol (SCC cell lines) (*Serrels et al., 2012*), (2) a primary breast tumor arising on the MMTV-PyMT genetically

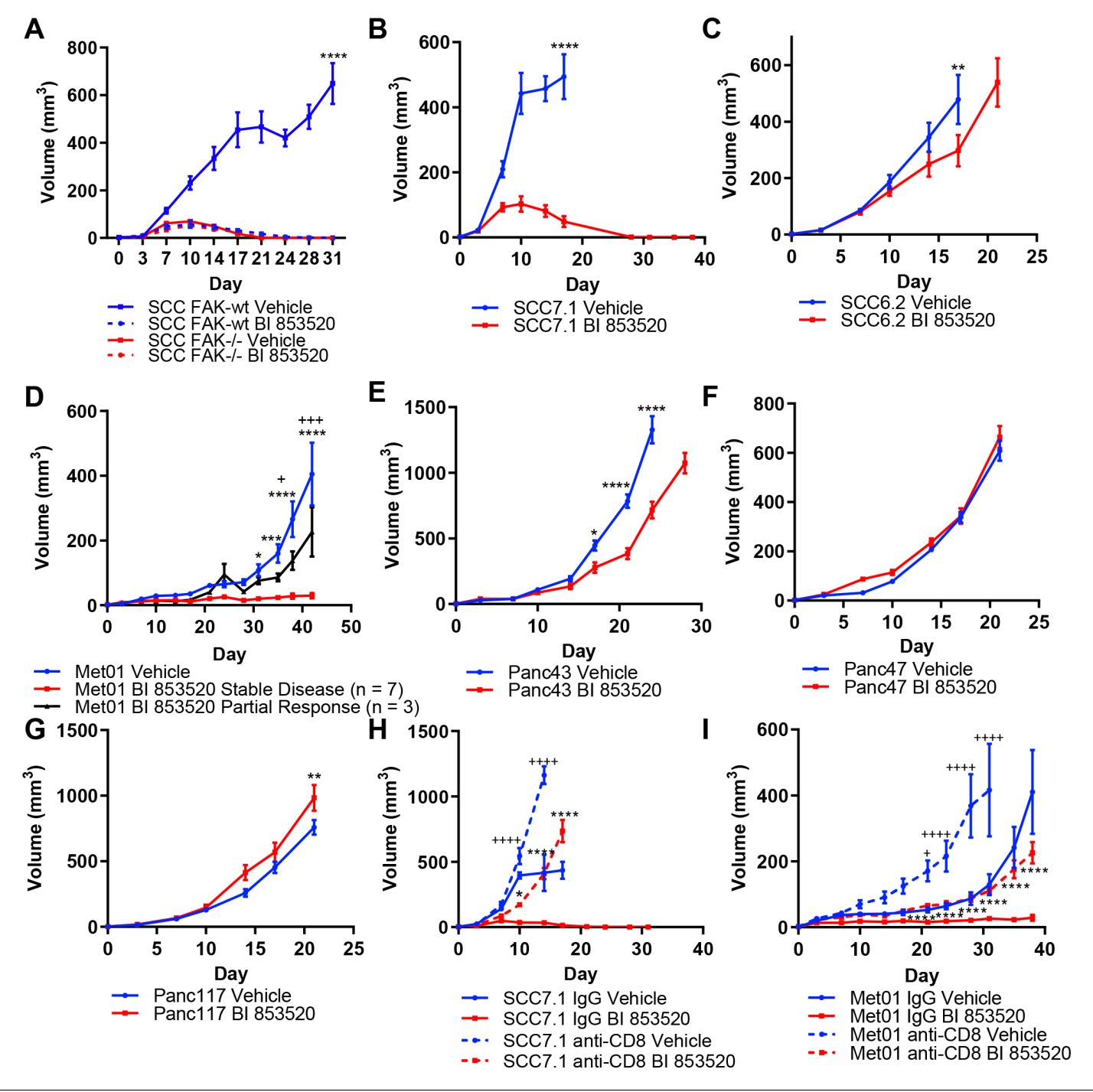

**Figure 1.** Treatment of a range of tumor models with the FAK kinase inhibitor BI 853520 identifies a spectrum of responses. (A - G) Representative graphs of tumor growth in immune-competent mice treated with either Vehicle or 50 mg/kg BI 853520. *=comparison of Vehicle to BI 853520, + = comparison of Vehicle to BI 853520 partial response in graph (D) n = 8–10 tumors per group. (H and I) Tumor growth of SCC7.1 and Met01 cells treated with either Vehicle or BI 853520 and Isotype control antibody (IgG) or anti-CD8 T-cell depleting antibody. + = comparison of IgG Vehicle to IgG BI 853520, *=comparison of anti-CD8 Vehicle to anti-CD8 BI 853520. * or + = p ≤ 0.05, ** or ++ = p ≤ 0.01, *** or +++ = p ≤ 0.001, **** or ++++ = p ≤ 0.0001, two-way ANOVA with Tukey's multiple comparison test. Data represented as mean + /- s.e.m. n = 6 tumors per group.

engineered mouse (GEM) model of breast cancer (Met01 cell line) (*Qian et al., 2011*), and (3) Pancreatic Ductal Adenocarcinoma (PDAC) arising on the *LSL-Kras$^{G12D/+}$;LSL-Trp53$^{R172H/+}$;Pdx1-Cre* (KPC) GEM model of pancreatic cancer (*Hingorani et al., 2005*) (Panc cell lines). FVB/N mice were injected subcutaneously with SCC7.1 or SCC6.2 cells and treated daily with either Vehicle or 50 mg/kg BI 853520. We observed complete regression of SCC7.1 tumors by day 27 (*Figure 1B*), while in contrast a different SCC cell line, SCC6.2, tumors exhibited only a modest growth delay (*Figure 1C*). Subcutaneous injection of Met01 cells into FVB/N mice followed by daily treatment with either Vehicle or BI 853520 identified a heterogeneous response to BI 853520, with 3 out of 10 tumors showing only a growth delay and 7 out of 10 tumors displaying stable disease for the duration of the study (*Figure 1D*). Lastly, subcutaneous injection of Panc43, Panc47, or Panc117 cells into C57BL/6 mice followed by daily treatment with either Vehicle or BI 853520 identified that Panc43 tumors exhibit only a modest growth delay in response to FAK inhibition (*Figure 1E*), while Panc47 and Panc117 tumors exhibit no response (*Figure 1F,G*). Collectively, these results identify a spectrum of response to treatment with BI 853520.

## CD8 T-cells mediate tumor regression/stable disease after BI 853520 treatment

Next we used antibody-mediated CD8 T-cell depletion to determine whether SCC7.1 tumor regression and Met01 stable disease was CD8 T-cell dependent. In mice treated with a CD8 T-cell depleting antibody, SCC7.1 and Met01 tumors exhibited only a modest growth delay in response to treatment with BI 853520 when compared to Vehicle treated controls, while in mice treated with an isotype control antibody all SCC7.1 tumors underwent complete regression and all Met01 tumors exhibited stable disease following treatment with BI 853520 (*Figure 1H,I*). Thus, FAK kinase inhibition acts via CD8 T-cell-mediated anti-tumor immunity to induce SCC7.1 tumor regression and Met01 stable disease.

## Expression of CD80 correlates with response to BI 853520

A number of FAK kinase inhibitors, including BI 853520, are now in early-phase (I/II) clinical trials as experimental cancer therapies (*de Jonge et al., 2019*; *Doi et al., 2019*; *Lee et al., 2015*; *Soria et al., 2012*; *Hirt et al., 2018*; *Shapiro et al., 2014*). However, we do not know which patients are most likely to benefit from these inhibitors, and to-date there are no validated biomarkers. Based on our findings in *Figure 1*, we hypothesized that either CD8 T-cell infiltration or expression of immune regulatory molecules that can modulate CD8 T-cell function might underlie sensitivity to FAK inhibition. To test this, we established subcutaneous tumors from each of the cell line models, and 12 days after implantation used flow cytometry to profile the CD8 T-cell infiltrate. This identified that SCC7.1 and Met01 tumors have a higher frequency of CD8 T-cell infiltration than SCC6.2, Panc43 and Panc47 tumors (*Figure 2—figure supplement 1A*). However, Panc117 tumors also had a higher frequency of CD8 T-cell infiltration when compared to SCC6.2, Panc43, and Panc47 tumors. Therefore, CD8 T-cell infiltration alone does not accurately predict response to BI 853520.

CD8 T-cell function can be regulated in a number of ways, including by the modulation of immune checkpoint pathways (*Marin-Acevedo et al., 2018*). We therefore sought to determine whether differential expression of these pathways correlated with response to BI 853520. Initially, we used flow cytometry to show that the immune checkpoint receptor Programmed Death Receptor 1 (PD-1) was expressed on activated (CD44$^+$) CD8 T-cells in all tumor types (*Figure 2—figure supplement 1B*). We next showed that none of the cell lines expressed PD-L1 or PD-L2 under basal culture conditions (not shown). However, when the cell lines were stimulated with 10 ng/ml interferon-gamma (IFNγ) for 24 hr all cell lines upregulated PD-L1 to a similar extent (*Figure 2—figure supplement 1C*), implying that all were capable of engaging the PD-1-PD-L1 axis. None of the cell lines upregulated the expression of PD-L2 in response to IFNγ treatment (not shown). Therefore, we did not identify any relationship between the PD-1-PD-L1 axis and response to BI 853520. PD-L1 and 2 belong to the family of B7 ligands that can engage coinhibitory and costimulatory receptors to regulate CD8 T-cell activity. We therefore profiled the expression of the B7 family of proteins across the tumor cell lines. While many of these ligands were not expressed by any of the cell lines, CD80 (also known as B7.1) was expressed exclusively by the SCC7.1 and Met01 cell lines that respond best to FAK inhibition (*Figure 2A,B*), thus correlating with response to BI 853520.

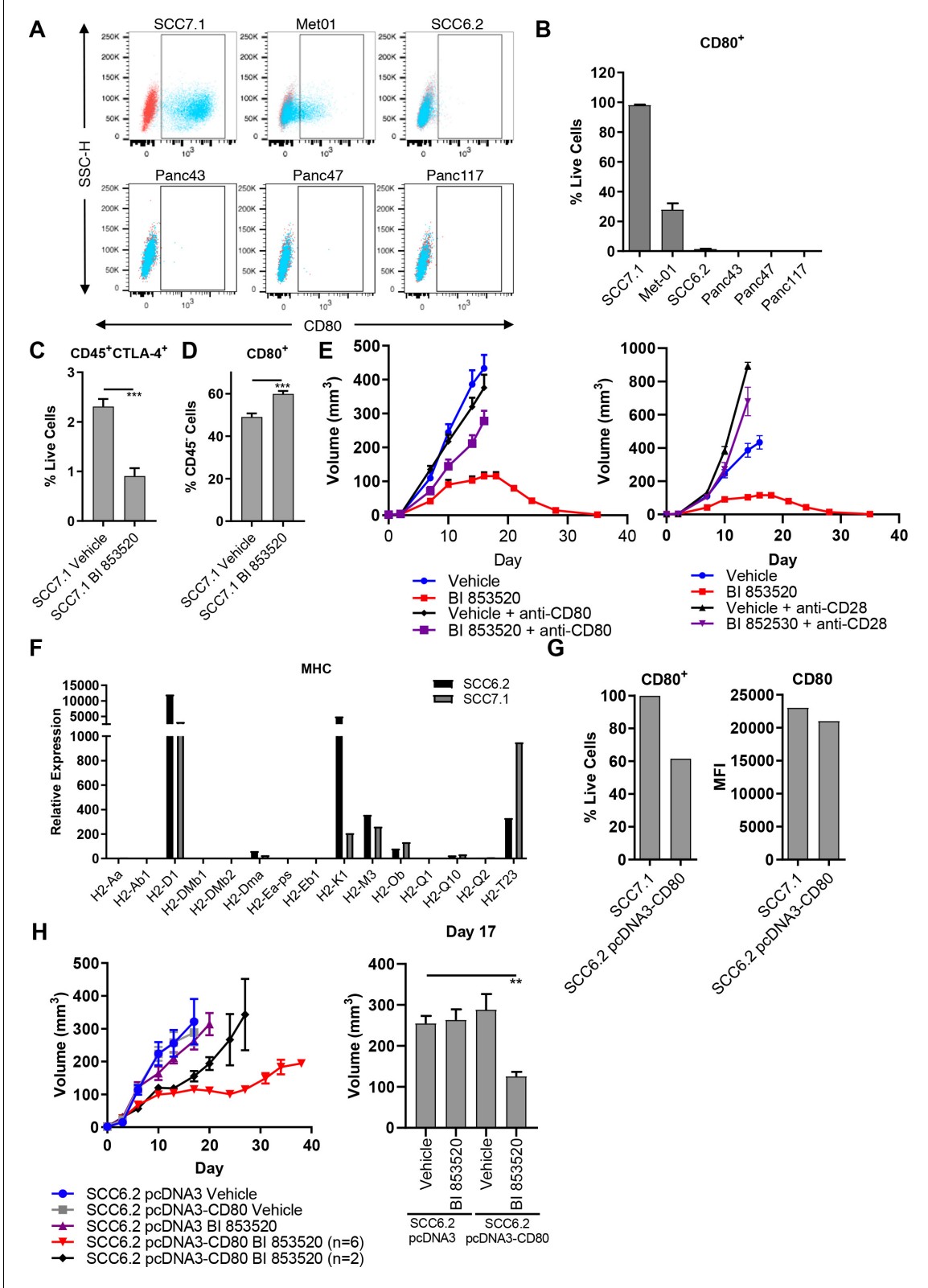

**Figure 2.** Expression of the immune costimulatory ligand CD80 renders tumors responsive to BI 853520. (**A**) Representative flow cytometry scatter plots of CD80 expression on SCC7.1, Met01, SCC6.2, Panc43, Panc47, and Panc117 cells under normal culture conditions. (**B**) Quantification of the percentage of live cells expressing CD80 in (**A**). Data represented as mean + /- s.e.m, n = 3. (**C**) Flow cytometry quantification of $CD45^+CTLA-4^+$ tumor infiltrating immune cells represented as a percentage of live cells. Data represented as mean + /- s.e.m. $p \leq 0.001$, two-tailed parametric unpaired t-test.

*Figure 2 continued on next page*

*Figure 2 continued*

(D) Flow cytometry quantification of CD45⁻CD80⁺ cells represented as a percentage of live cells. Data represented as mean + / - s.e.m. p≤0.001, two-tailed parametric unpaired t-test. (E) *Left* - Subcutaneous tumor growth of SCC7.1 cells treated with either Vehicle or 50 mg/kg BI 853520 ± 100 μg anti-CD80 antibody. *Right* – subcutaneous tumor growth of SCC7.1 cells treated with either Vehicle or 50 mg/kg BI 853520 ± 100 μg anti-CD28 antibody. Data represented as mean + / - s.e.m. Vehicle (blue) and BI 853520 (red) datasets are the same for both graphs. (F) Nanostring gene expression analysis of Major Histocompatibility Complex (MHC) genes in SCC7.1 and SCC6.2 cells. (G) Comparison of CD80 surface expression between SCC7.1 cells and SCC6.2 cells transfected with a pcDNA3-CD80 vector. *Left* - Flow cytometry analysis of the percentage of cells expressing CD80. *Right* - mean fluorescent intensity (MFI) of CD80 cell surface expression. (H) *Left* - Subcutaneous tumor growth of SCC6.2 cells transfected with either pcDNA3 empty vector or pcDNA3-CD80 vector and treated with either Vehicle or 50 mg/kg BI 853520. *Right* – Comparison of tumor volume on day 17 post-implantation of tumor cells. Data represented as mean + / - s.e.m. p≤0.01, one-way ANOVA with Dunnett's multiple comparison. n = 8–10 tumors per group.

The online version of this article includes the following figure supplement(s) for figure 2:

**Figure supplement 1.** Regulation of PD-L1/PD-1 axis does not correlate with response of tumors to BI 853520.

## BI 853520 alters the balance of CD80: CTLA-4

CD80 is a ligand for two receptors, CTLA-4 and CD28 (*Linsley et al., 1991*; *Linsley et al., 1990*). CTLA-4 is a co-inhibitory receptor associated with suppression of anti-tumor immune responses and has a higher affinity than CD28 for binding CD80 (*van der Merwe et al., 1997*). Through the process of trans-endocytosis CTLA-4 can deplete CD80 from the surface of neighboring cells (*Qureshi et al., 2011*), preventing CD80 - CD28 interaction and resulting T-cell co-stimulation. Regulatory T-cells, which we have previously shown are depleted from the tumor environment in response to treatment with a FAK inhibitor (*Serrels et al., 2015*), represent a major cellular source of CTLA-4 (*Walker and Sansom, 2015*). We therefore postulated that treatment with BI 853520 may result in a reduction in CTLA-4 expressing cells within the TME and a concomitant increase in CD80 availability. To test this, FVB/N mice were injected subcutaneously with SCC7.1 cells and treated daily with either Vehicle or 50 mg/kg BI 853520. Mice were culled 12 days post-tumor cell implantation and tumors processed for analysis by flow cytometry (*Supplementary file 1*). Treatment with BI 853520 resulted in a significant decrease in CTLA-4⁺ immune cells (*Figure 2C*) and an increase in the frequency of CD45⁻ cells expressing CD80 on their cell surface (*Figure 2D*). Therefore, treatment with a FAK inhibitor shifts the balance in favor of CD80 - CD28 interaction and T-cell co-stimulation.

## CD80/CD28 blockade impairs the anti-tumor efficacy of BI 853520

To determine whether the CD80 - CD28 pathway was actually involved in regulating response to BI 853520, mice were injected subcutaneously with SCC7.1 cells and treated daily with either Vehicle, Vehicle + 100 μg anti-CD80 or 100 μg anti-CD28, 50 mg/kg BI 853520, or 50 mg/kg BI 853520 + 100 μg anti-CD80 or 100 μg anti-CD28. Both CD80 and CD28 blockade significantly impaired the anti-tumor efficacy of BI 853520 (*Figure 2E*), implying an important role for CD80 in regulating response to a FAK inhibitor.

## Enforced expression of CD80 renders tumors responsive to FAK inhibition

To further investigate the relationship between CD80 expression on cancer cells and sensitivity of tumors to the BI 853520 FAK inhibitor, we next expressed CD80 into the BI 853520-resistant SCC6.2 cell line (*Figure 1B*). Using nanostring gene expression analysis we first confirmed that SCC6.2 cells have a similar expression profile of Major Histocompatibility Complex (MHC) molecules to that of CD80⁺ SCC7.1 cells that respond to BI 853520 (*Figure 2F*), ensuring that SCC6.2 cells express MHC-I molecules required for CD8 T-cell recognition and effective anti-tumor immunity. We next transfected SCC6.2 cells with either the mammalian expression vector pcDNA3, or pcDNA3-CD80 (*Figure 2G*), and injected the cells subcutaneously into FVB/N mice and monitored tumor growth in response to either treatment with Vehicle or BI 853520 (*Figure 2H*). Expression of CD80 had no impact on SCC6.2 tumor growth. However, SCC6.2 pcDNA3-CD80 tumors showed a significantly improved response to BI 853520 in comparison to either Vehicle treated controls or SCC6.2 pcDNA3 tumors treated with BI 853520. Thus, CD80 expression sensitizes SCC6.2 tumors to BI 853520. Collectively, these findings suggest that cancer cell expression of CD80, most likely in combination with

MHC-I expression, has the potential to identify tumors with increased sensitivity to FAK kinase inhibition.

## CD80+ SCC cells co-express genes associated with a cancer stem cell phenotype previously reported in murine and human SCC

CD80 is most notably expressed by antigen presenting cells. However, the prevalence of CD80 expression in human cancer cells has not been extensively characterized. Recent studies suggest that CD80 expression is upregulated in human colonic epithelial cells isolated from preneoplastic adenomas (*Marchiori et al., 2019*; *Scarpa et al., 2015*), while in skin SCC, a population of Cancer Stem Cells (CSCs) that express CD80 have been reported in both murine and human tumors (*Miao et al., 2019*). These murine skin CSCs are identified by expression of a number of genes including *Cd34*, *Itga6*, high expression of integrins in general, and responsiveness to TGFβ as indicated by activation of downstream signaling including expression of *Cdkn1a* (*Miao et al., 2019*; *Oshimori et al., 2015*; *Schober and Fuchs, 2011*). Furthermore, previous related studies have also shown that CD34+ skin CSCs reside near the tumor vasculature at the tumor stromal interface and express *Vegfa* (*Beck et al., 2011*). We therefore sought to determine whether the CD80+ SCC7.1 cells that respond to BI 853520 when grown as tumors had any resemblance to the CD80+ skin CSCs that have been previously reported (*Miao et al., 2019*). Nanostring gene expression analysis and flow cytometry identified that SCC7.1 cells express significantly more *Cd34* and *Itga6* (Integrin alpha-6) than SCC6.2 cells (*Figure 3A,B*). Furthermore, SCC7.1 cells also expressed higher levels of *Itga2*, *Itgb1*, *Itgb2*, *Itgb4*, *Cdkn1a*, and *Vegfa* when compared to SCC6.2 cells (*Figure 3C–E*). Thus, SCC7.1 cells express a gene signature that overlaps with that of CD80+ CSCs previously identified in murine and human SCCs.

## CD80 is expressed by a broad range of human cancer cells

Attempts to more broadly define the extent of cancer cell CD80 expression using bulk tumor transcriptomics data, for example using the Cancer Genome Atlas (TCGA), are confounded by the fact that CD80 is also expressed by infiltrating antigen presenting cells. Therefore, we used publicly available transcriptomics data spanning a large number of human cancer cell lines available via the Human Cancer Cell Line Encyclopedia (*Ghandi et al., 2019*). This identified that *CD80* transcript is present in a substantial proportion of cell lines from solid epithelial cancers, albeit a considerable proportion were also negative for *CD80* expression (*Figure 4A*). *PTK2* (FAK gene) was highly expressed in all cell lines originating from solid epithelial cancers (*Figure 4B*).

Interestingly, although perhaps not unexpected, this analysis also identified that hematological malignancies including lymphomas generally expressed higher levels of *CD80* when compared to cell lines originating from solid epithelial cancers (*Figure 4A*). *PTK2* was only expressed in a subset of cell lines from hematological malignancies (*Figure 4B*). Plotting *CD80* expression against *PTK2* expression further confirmed these findings (*Figure 4C*). Given that hematological malignancies represent a broad group of cancers we also determined whether co-expression of high *CD80* and high *PTK2* was restricted to particular hematological malignancies by subdividing this category based on origin (*Figure 4D,E*). These analyses suggested that co-expression of high levels of *CD80* and *PTK2* transcript occurs preferentially in cancer cell lines originating from patients with Burkitt's Lymphoma, Hodgkin's Lymphoma, Chronic Myeloid Leukemia, and Diffuse Large B-cell (DLBCL) Lymphoma. To our knowledge FAK inhibitors have not been tested in patients diagnosed with these cancer types. Collectively, these results suggest that CD80-expressing cancer cells are present in some solid epithelial cancers and some blood cancers, supporting the potential to select patients based on cancer cell CD80 expression for treatment with a FAK inhibitor.

## BI 853520 enhances the effects of T-cell costimulatory receptor agonistic antibodies

Given that not all cancer cell lines expressed CD80, we next considered the concept of T-cell co-stimulation in combination with FAK inhibition using a number of agonistic antibodies targeting T-cell costimulatory receptors, including glucocorticoid-induced TNFR-related protein (GITR, also known as Tumor necrosis factor receptor superfamily member 18 (TNFRSF18)), CD40 (also known as Tumor necrosis factor receptor superfamily member 5 (TNFRSF5)), 4-1BB (also known as tumor

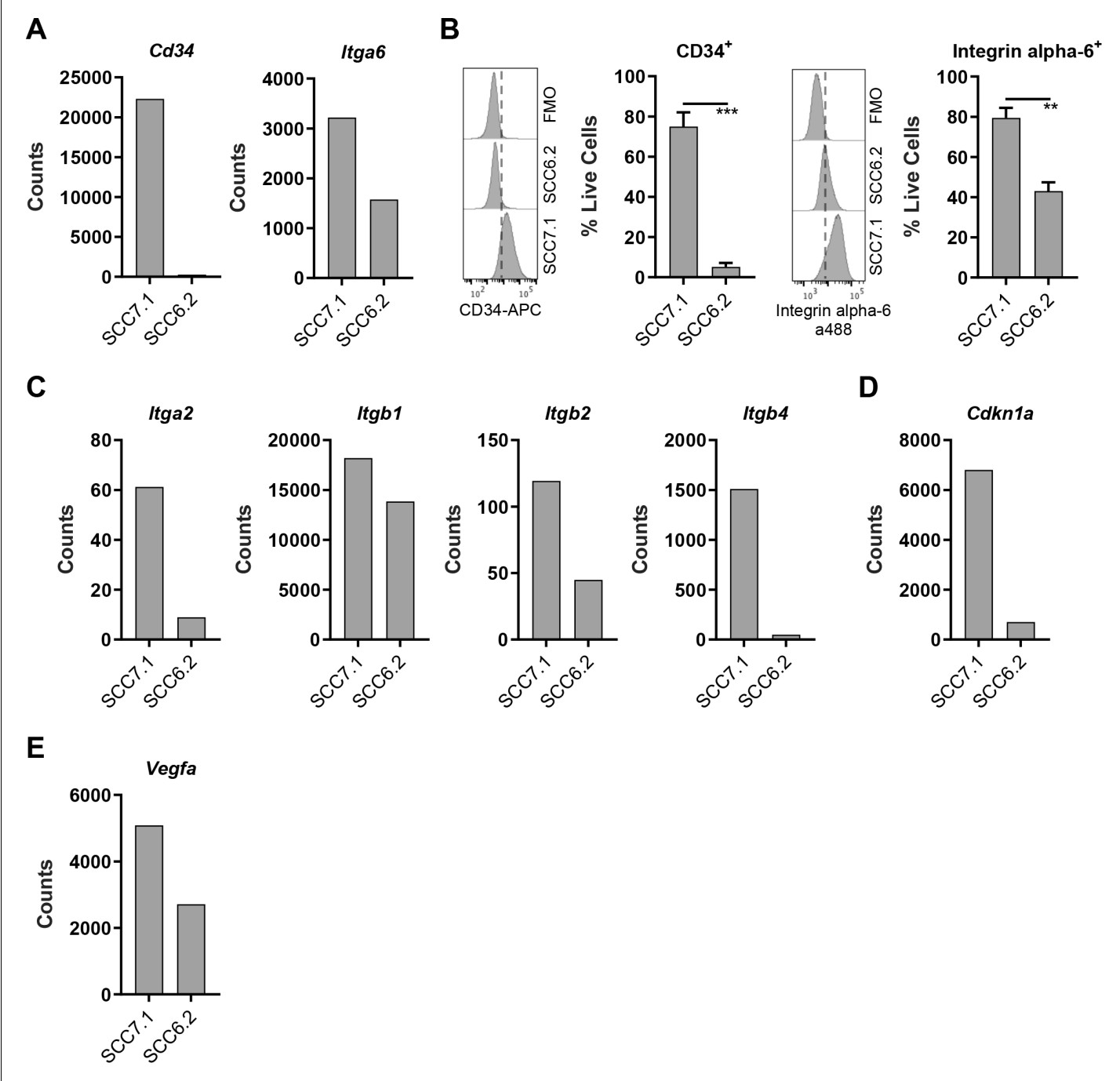

**Figure 3.** CD80[+] SCC cells express markers associated with Cancer Stem Cells. (**A**) Nanostring gene expression analysis using RNA isolated from SCC7.1 and SCC6.2 cells. (**B**) *Histograms* - representative flow cytometry histograms of fluorescent intensity in SCC7.1 and SCC6.2 cells stained with a combination of anti-CD34 APC and anti-integrin alpha-6 alexa488 conjugated antibodies. FMO control represented full stain minus the antibody of interest. *Graphs* – flow cytometry quantification of the percentage of live cells expressing either CD34 or integrin alpha-6. (**C – E**) Nanostring gene expression analysis using RNA isolated from SCC7.1 and SCC6.2 cells. Nanostring data normalised using nSolver software and represented as reporter probe counts. Flow cytometry data represented as mean + /- s.e.m. ***p≤0.001, **p≤0.01, unpaired two-tailed t-test.

The online version of this article includes the following source data for figure 3:

**Source data 1.** Nanostring gene expression analysis of SCC7.1 and SCC6.2 cells.

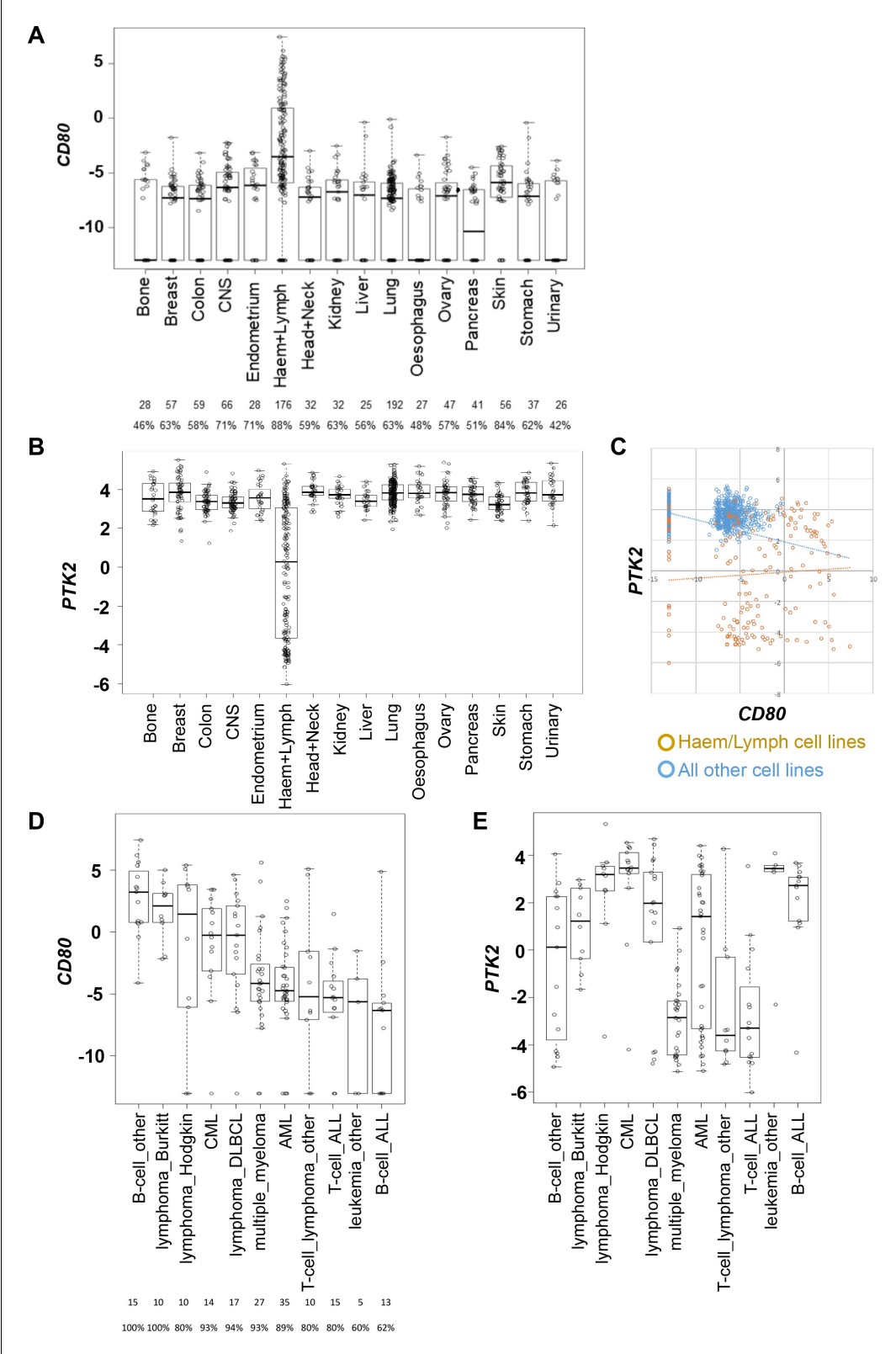

**Figure 4.** *CD80* and *PTK2* expression in human cancer cell lines. (**A**) Analysis of *CD80* expression in RNAseq datasets derived from human cancer cell lines. The number of cell lines from each tumor type is listed below along with the percentage positive for expression. (**B**) Analysis of *PTK2* expression in the same datasets from human cancer cell lines. (**C**) Scatter plot of *CD80* expression against *PTK2* expression in all cell lines. (**D**) Analysis of *CD80*

*Figure 4 continued*

expression in human cancer cell lines from different types of hematological malignancies. (E) Analysis of *PTK2* expression in human cancer cell lines from different hematological malignancies.

necrosis factor receptor superfamily member 9 (TNFRSF9) and CD137), and OX40 (also known as Tumor necrosis factor receptor superfamily member 4 (TNFRSF4) and CD134). To investigate the potential anti-tumor efficacy of the FAK inhibitor in combination with these antibodies, SCC6.2 cells were injected subcutaneously into FVB/N mice and mice treated with either Vehicle, Vehicle + 100 µg/mouse anti-GITR, Vehicle + 100 µg/mouse anti-CD40, Vehicle + 100 µg/mouse anti-4-1BB, Vehicle + 100 µg/mouse anti-OX40, BI 853520 + 100 µg/mouse anti-GITR, BI 853520 + 100 µg/mouse anti-CD40, BI 853520 + 100 µg/mouse anti-4-1BB or BI 853520 + 100 µg/mouse anti-OX40 (treatment schedule shown in *Figure 5A*). Treatment of SCC6.2 tumors with Vehicle + anti-GITR agonistic antibody resulted in a small growth delay when compared to control treated tumors, which was not improved when the anti-GITR antibody was combined with BI 853520 (*Figure 5B*). Similar results were observed using an anti-CD40 agonistic antibody alone or in combination with BI 853520 (*Figure 5C*). However, treatment of SCC6.2 tumors with Vehicle + anti-4-1BB agonistic antibody resulted in 2 out of 8 tumors undergoing complete regression and the remaining six tumors exhibiting a delay in growth when compared to control treated tumors. Combining anti-4-1BB with BI 853520 further improved this response, resulting in 4 out of 8 tumors undergoing complete regression, and the remaining four tumors exhibiting an improved growth delay when compared with Vehicle + anti-4-1BB (*Figure 5D*). Treatment of SCC6.2 tumors with Vehicle + anti-OX40 agonistic antibody resulted in 2 out of 8 tumors undergoing complete regression, and the remaining six exhibiting a growth delay when compared to control treated tumors. Combining anti-OX40 with BI 853520 resulted in complete regression of all tumors (*Figure 5E*). Thus, the FAK kinase inhibitor in combination with activation of either OX40 or 4-1BB represents a potentially promising therapeutic strategy. To determine whether the potent anti-tumor efficacy of BI 853520 + anti-OX40 was dependent on CD8 T-cells, we depleted CD8 T-cells using an anti-CD8 antibody, and measured SCC6.2 tumor growth in FVB/N mice receiving either Vehicle or BI 853520 + anti-OX40 (*Figure 5F*). In mice receiving anti-CD8 antibody the combination of BI 853520 + anti-OX40 resulted in a growth delay when compared to mice treated with anti-CD8 antibody and Vehicle. However, only in mice receiving an isotype control antibody the combination of BI 853520 + anti-OX40 resulted in complete regression of all tumors, implying that tumor regression was dependent on CD8 T-cells. An anti-tumor CD8 T-cell response should also result in lasting immunological memory. We therefore aged mice for two months following regression of SCC6.2 tumors in response to treatment with BI 853520 + anti-OX40, and rechallenged these mice with a fresh preparation of SCC6.2 cells. All mice remained tumor free during the two months following initial regression of SCC6.2 tumors and following rechallenge no tumor growth was observed (*Figure 5G*). In contrast, subcutaneous injection of the same SCC6.2 cell preparation into FVB/N mice that had never previously been challenged resulted in robust growth of all tumors. Therefore, SCC6.2 tumor regression in response to BI 853520 + anti-OX40 resulted in lasting immunological memory that renders mice resistant to further tumor growth.

## BI 853520 and anti-OX40 have overlapping and distinct activity

To define the mechanism underpinning the improved efficacy of BI 853520 in combination with anti-OX40, we next profiled the tumor immune cell infiltrate in response to treatment with either Vehicle, anti-OX40, BI 853520, or BI 853520 + anti-OX40. SCC6.2 cells were injected subcutaneously into FVB/N mice and treatment administered as detailed in *Figure 5A*. Mice were culled 12 days post tumor cell implantation and tumors processed for analysis using flow cytometry (*Figure 6—figure supplements 1* and *2*; *Supplementary files 1*, *2*, *3*, and *4*). Both anti-OX40 and BI 853520 treatment resulted in a reduction in Tregs, as did the combination of BI 853520 + anti-OX40 (*Figure 6A*). In tumors treated with anti-OX40 or anti-OX40 + BI 853520, this was accompanied by an increase in the frequency of non-Treg CD4$^+$ T-cells (*Figure 6B*). Treatment with anti-OX40 or BI 853520 + anti-OX40 also resulted in an increase in effector CD8 T-cells (CD8$^{eff}$) when compared to control treated and BI 853520 treated tumors (*Figure 6C*), implying that anti-OX40 is required to drive increased

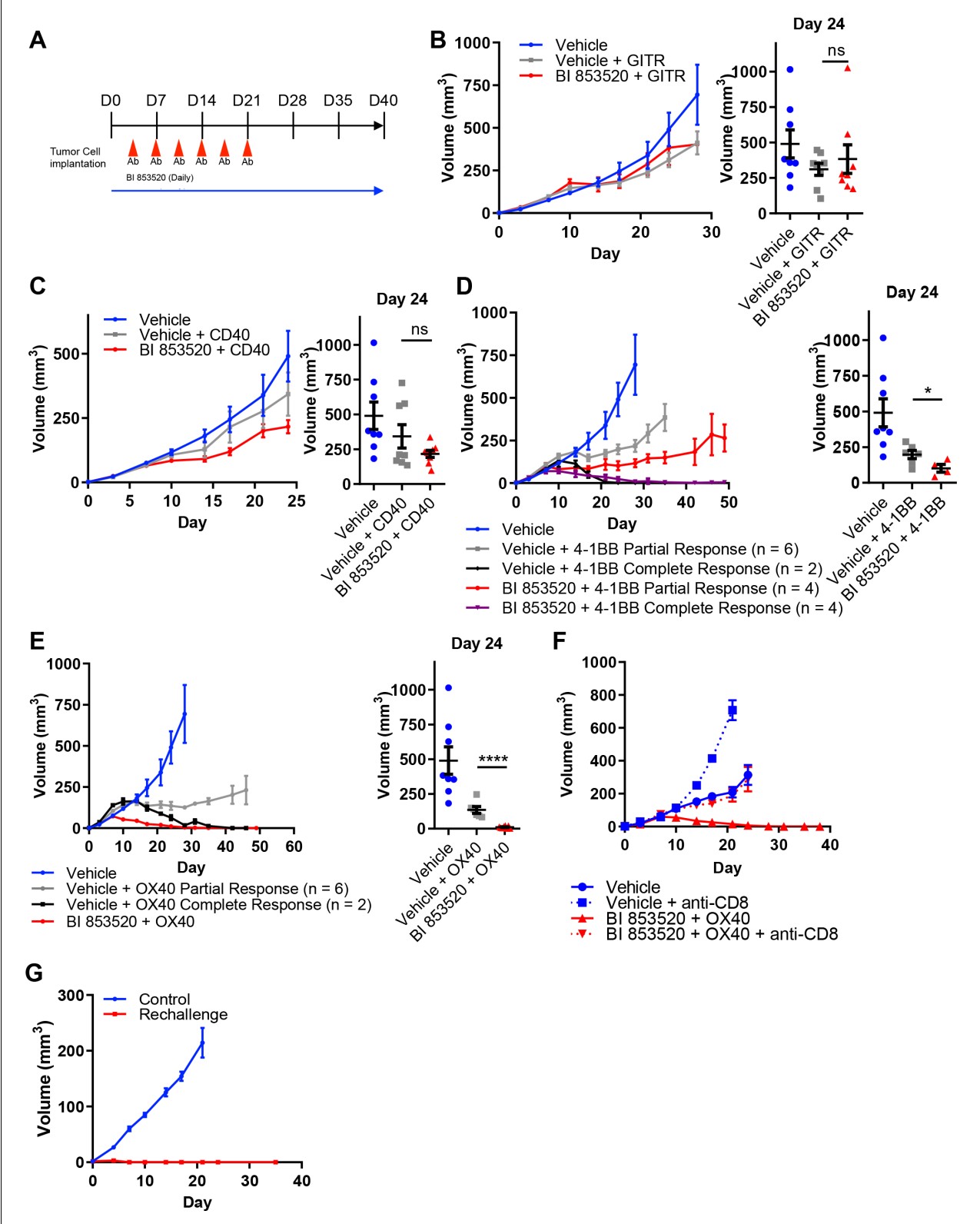

**Figure 5.** BI 853520 improves the response of SCC6.2 tumors to 4-1BB and OX40 agonistic antibodies. (**A**) Dosing schedule. (**B – E**) *Left* - Subcutaneous growth of SCC6.2 tumors treated with either Vehicle or 50 mg/kg BI 853520 in combination with either 100 μg GITR, CD40, 4-1BB, or OX40 agonistic antibodies. *Right* - Comparison of tumor volume on day 24 post-implantation of tumor cells. Graph represents individual tumor measurements together with the mean + /- s.e.m. (**F**) Subcutaneous growth of SCC6.2 tumors treated with either Vehicle or BI 853520 + OX40 in combination with either isotype
*Figure 5 continued on next page*

*Figure 5 continued*

control or anti-CD8 T-cell depleting antibodies. (G) Subcutaneous growth of SCC6.2 tumors implanted into either naïve FVB mice (Control) or FVB mice from E in which treatment with BI 853520 + OX40 resulted in complete tumor regression and no tumor regrowth over a 2 month period off treatment (Rechallenge). (B – E) ns = not significant, *=p ≤ 0.05, ****=p ≤ 0.0001, unpaired t-test comparing immunotherapy to immunotherapy + BI 853520. Data represented as mean + /- s.e.m. n = 8–10 tumors per group.

CD8$^{eff}$ T-cell numbers in SCC6.2 tumors. However, this did not explain why BI 853520 enhanced the anti-tumor immune response when used in combination with anti-OX40. We found no statistically significant difference in the frequency of CD8 T-cells co-expressing either PD-1 and LAG-3 or PD-1 and Tim-3 (*Figure 6—figure supplement 3*), although there was a trend towards increased PD-1$^+$Tim-3$^+$ CD8 T-cells in both OX40 and OX-40 + BI 853520 treated tumors. OX40 treatment has been reported to increase expression of ICOS on both CD4 and CD8 T-cells, and modulating this pathway can impact the therapeutic response to anti-OX40 antibodies in mouse models of cancer (*Metzger et al., 2016*). Treatment of SCC6.2 tumors with anti-OX40 resulted in an increase in ICOS expression on both non-Treg CD4 T-cells and CD8$^{eff}$ T-cells, and this was further increased, especially on CD8$^{eff}$ T-cells, when anti-OX40 was used in combination with BI 853520 (*Figure 6D*; *Figure 6—figure supplement 4*). Interestingly, increased expression of ICOS on CD8$^{eff}$ T-cells was also observed in response to BI 853520 treatment in SCC7.1 tumors that express the co-stimulatory molecule CD80 (*Figure 6—figure supplement 5*), suggesting that FAK inhibition may more broadly regulate expression of ICOS on CD8 T-cells when a T-cell co-stimulatory signal is present.

Further immune profiling did not find significant differences in the frequency of macrophages, granulocytic myeloid-derived suppressor cells (G-MDSC), monocytic myeloid-derived suppressor cells (M-MDSC), or cancer cells (*Figure 6E*; *Figure 6—figure supplement 6*). However, we did observe a significant increase in CD11b$^+$dendritic cells (DCs) in response to OX40, which was restored to control levels when OX40 was combined with BI 853520 (*Figure 6—figure supplement 6D*). Furthermore, we found that macrophage expression of the immune checkpoint ligand PD-L1 and was not altered (*Figure 6F*), but that BI 853520 either alone or in combination with OX40 resulted in a decrease in PD-L2 expression by macrophages (*Figure 6G*). Comparison of PD-L2 expression on cancer cells, G-MDSCs, M-MDSCs, macrophages, and CD11b+ DCs revealed that macrophages and DCs express the highest levels of PD-L2 from this range of cell types (*Figure 6H*). BI 853520 treatment either alone or in combination with anti-OX40 also resulted in a reduction in PD-L2$^+$ cancer cells and M-MDSCs (*Figure 6I*). Therefore, BI 853520 treatment generally depletes the availability of PD-L2 within the tumor microenvironment, likely impacting on decrease of PD-L2-PD-1 signaling and ultimately escape of CD8 T-cells from exhaustion. To ascertain whether treatment with a FAK inhibitor could directly regulate expression of PD-L2 on macrophages, in vitro bone marrow-derived macrophages were treated with 100 nM BI 853520 and stimulated with 10 ng/ml interleukin-4 (IL-4) (*Figure 6—figure supplement 7*). BI 853520 treatment resulted in a small decrease in PD-L2 expression following stimulation with IL-4, however this did not appear to fully account for the substantial reduction observed in vivo (*Figure 6G*) suggesting that FAK-dependent regulation of PD-L2 expression in the TME may be more complex.

## PD-L2 and ICOS contribute to the efficacy of BI 853520 + anti-OX40/4–1 BB

To investigate whether regulation of PD-L2 by BI 853520 might contribute to enhanced efficacy when used in combination with either anti-OX40 or anti-4-1BB, SCC6.2 tumors were implanted into FVB/N mice and treated with either anti-OX40, anti-OX40 + anti-PD-L2, anti-41BB, or anti-4-1BB + anti-PD-L2 (*Figure 6J,K*). By day 21, 5 out of 10 tumors treated with anti-OX40 + anti-PD-L2 had undergone complete regression, while in contrast all tumors treated with anti-OX40 alone were still present, albeit small. Anti-PD-L2 in combination with anti-4-1BB resulted in a small decrease in the mean tumor volume at day 21 when compared to anti-4-1BB alone. However, this was not statistically significant. Therefore, regulation of PD-L2 likely contributes more towards the anti-tumor efficacy of BI 853520 when used in combination with anti-OX40.

Similar experiments were performed to investigate the role of ICOS in the anti-tumor response to either BI 853520 + anti-OX40 or BI 853520 + anti-4-1BB. Signaling through ICOS has been reported to enhance the anti-tumor efficacy of OX40 (*Metzger et al., 2016*). We therefore used an ICOS-

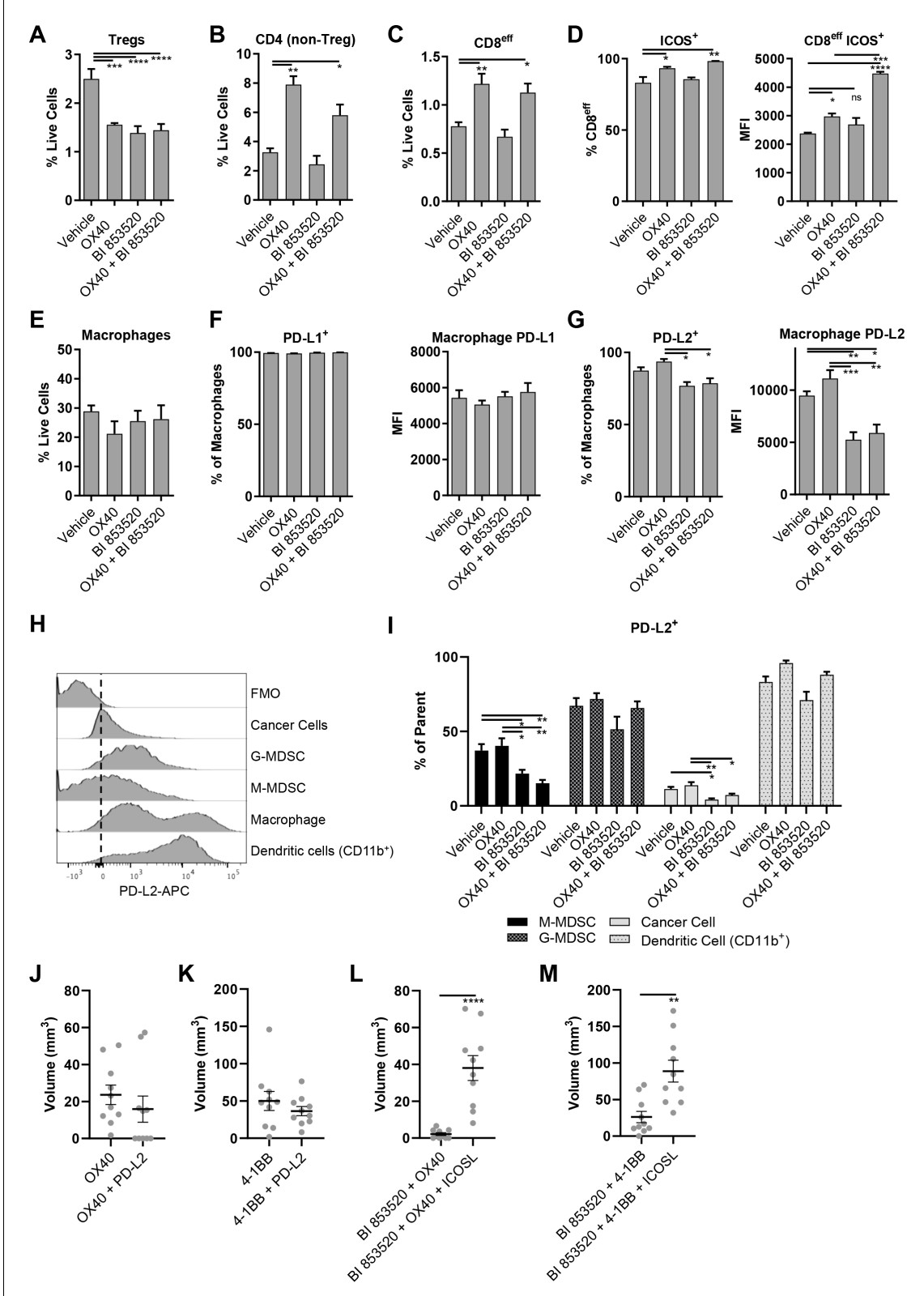

**Figure 6.** OX40 and BI 853520 display overlapping and distinct immune modulatory activity. (**A–C**) Flow cytometry quantification of tumor infiltrating Tregs, CD4 (non-Treg) T-cells, and CD8$^{eff}$ T-cells. (**D**) *Left* – Flow cytometry quantification of the percentage of CD8$^{eff}$ T-cells expressing ICOS. *Right* - Flow cytometry quantification of the median fluorescent intensity (MFI) of ICOS expression on CD8$^{eff}$ T-cells. (**E**) Flow cytometry quantification of tumor infiltrating macrophages as a percentage of live cells. (**F**) *Left* - Flow cytometry quantification of the percentage of macrophages expressing PD-L1. *Figure 6 continued on next page*

*Figure 6 continued*

Right - mean fluorescent intensity of PD-L1 expression on macrophages. (G) *Left* - Flow cytometry quantification of the percentage of macrophages expressing PD-L2. *Right* - mean fluorescent intensity of PD-L2 expression on macrophages. (H) Representative histogram of PD-L2 fluorescent intensity in cancer cells, G-MDSCs, M-MDSCs, Macrophages, and CD11b+Dendritic cells from a control sample stained with all antibodies. FMO is a fully stained samples except for PD-L2. (I) Flow cytometry quantification of the percentage of M-MDSCs, G-MDSCs, cancer cells, and dendritic cells positive for expression of PD-L2. (J) Comparison of subcutaneous tumor volume 21 days post-implantation of SCC6.2 cells. Tumors treated with either anti-OX40 or anti-OX40 + anti-PD-L2. (K) Comparison of subcutaneous tumor volume 21 days post-implantation of SCC6.2 cells. Tumors treated with either anti-4-1BB or anti-4-1BB + anti-PD-L2. (L) Comparison of subcutaneous tumor volume 21 days post-implantation of SCC6.2 cells. Tumors treated with either anti-OX40 or anti-OX40 + anti-ICOSL. (M) Comparison of subcutaneous tumor volume 21 days post-implantation of SCC6.2 cells. Tumors treated with either anti-4-1BB or anti-4-1BB + anti-ICOSL. (A - I) Data represented as mean + /- s.e.m. *=p $\leq$ 0.05, **=p $\leq$ 0.01, ***=p $\leq$ 0.001, ****=p $\leq$ 0.0001, ordinary one-way ANOVA with Tukey's multiple comparison. n = 4–8 tumors per treatment condition. (J - M) Data represented as individual tumor measurements together with the mean + /- s.e.m. ***=p $\leq$ 0.001, unpaired nonparametric Mann-Whitney test. Dosing schedule identical to *Figure 5A*. n = 10 tumors per group.

The online version of this article includes the following figure supplement(s) for figure 6:

**Figure supplement 1.** Flow cytometry T-cell gating strategy.

**Figure supplement 2.** Flow cytometry myeloid cell gating strategy.

**Figure supplement 3.** Expression of markers associated with T-cell exhaustion of CD8[eff] T-cells in SCC6.2 tumors treated with either Vehicle, OX40, BI 853520 or OX40 + BI 853520.

**Figure supplement 4.** OX40 and OX40 + BI 853520 treatment enhances ICOS expression on CD4 non-Treg cells in SCC6.2 tumors.

**Figure supplement 5.** BI 853520 treatment enhances ICOS expression on CD8[eff] T-cells in SCC7.1 tumors.

**Figure supplement 6.** Frequency of immune cell populations in SCC6.2 tumors treated with either Vehicle, OX40, BI 853520 or OX40 + BI 853520.

**Figure supplement 7.** BI 853520 partially inhibits macrophage expression of PD-L2 following stimulation with IL4.

ligand (ICOSL) blocking antibody to inhibit ICOS signaling. Blocking ICOSL had a significant impact on the anti-tumor efficacy of both BI 853520 + anti-OX40 and BI 853520 + anti-4-1BB (*Figure 6L,M*), suggesting that enhanced T-cell expression of ICOS may be an important mechanism through which a FAK inhibitor can potentiate the anti-tumor activity of anti-OX40 and anti-4-1BB targeted therapies.

## BI 853520 renders Panc47 tumors responsive to immune costimulatory antibodies

SCC6.2 cancer cells were generated using the DMBA/TPA chemical carcinogenesis protocol that results in a large number of somatic mutations (*Nassar et al., 2015*), and therefore likely increased immunogenicity. Hence, one would predict that these will respond better to immunotherapy than poorly immunogenic tumor models such as those derived from the KPC GEM model of pancreatic cancer. We therefore tested the same combinations of BI 853520 + /- immune costimulatory antibodies to determine whether a FAK inhibitor combined with T-cell co-stimulation could also influence a more poorly immunogenic tumor model. Panc47 cells were injected subcutaneously into C57BL/6 mice and mice treated with either Vehicle or BI 853520 + /- immune costimulatory antibodies (treatment schedule shown in *Figure 5A*). Anti-GITR alone had no effect on the growth of Panc47 tumors, while in combination with BI 853520 we observed a small growth delay (*Figure 7A*). In contrast, treatment with an anti-CD40 agonistic antibody resulted in a significant delay in the growth of Panc47 tumors, and this was not further enhanced when used in combination with BI 853520 (*Figure 7B*). Treatment of Panc47 tumors with an anti-4-1BB agonistic antibody had no effect on tumor growth, but a significant growth delay was observed when used in combination with BI 853520 (*Figure 7C*). Treatment with an anti-OX40 agonistic antibody had no effect on Panc47 tumor growth, but when used in combination with BI 853520 caused a significant growth delay (*Figure 7D*). Therefore, a FAK kinase inhibitor can render Panc47 tumors sensitive to anti-4-1BB and anti-OX40 immunotherapies, further supporting the development of these combinations as potential cancer therapies.

In contrast to the response of SCC6.2 tumors to the combination of BI 853520 + anti-OX40 or BI 853520 + anti-4-1BB, we did not observe regression of Panc47 tumors when using either of these combinations. A FAK inhibitor, VS-4718, was previously found to sensitize pancreatic tumors to immune checkpoint therapy, specifically anti-PD-1 and anti-CTLA-4, when used in combination with the chemotherapy Gemcitabine (*Jiang et al., 2016*). We therefore tested Gemcitabine alone and in combination with either BI 853520, anti-OX40, or BI 853520 + anti-OX40. Panc47 cells were injected

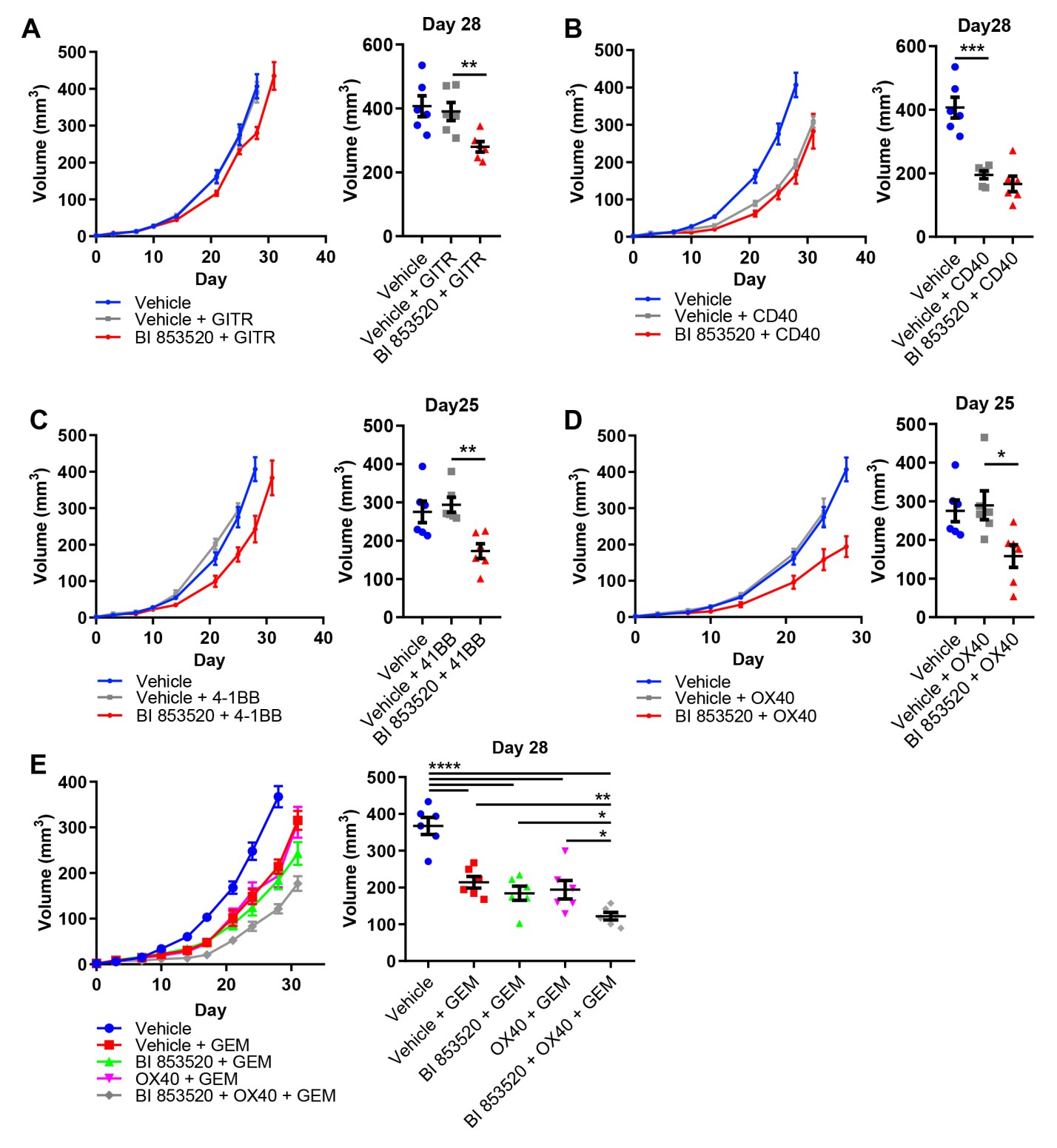

**Figure 7.** BI 853520 renders Panc47 tumors responsive to 4-1BB and OX40 agonistic antibodies. (A – D) Left - Subcutaneous growth of Panc47 tumors treated with either Vehicle or 50 mg/kg BI 853520 in combination with either 100 μg GITR, CD40, 4-1BB, or OX40 agonistic antibodies. Right - Comparison of tumor volume on day 25/28 post-implantation of tumor cells. Graph represents individual tumor measurements together with the mean + /- s.e.m. Dosing schedule identical to *Figure 5A*. (E) Left - Subcutaneous growth of Panc47 tumors treated with either Vehicle, Vehicle + 100 mg/kg Gemcitabine (GEM), 50 mg/kg BI 853520 + GEM, OX40 + GEM, or BI 853520 + OX40 + GEM. Right - Comparison of tumor volume on day 28 post-
*Figure 7 continued on next page*

*Figure 7 continued*

implantation of tumor cells. Graph represents individual tumor measurements together with the mean + /- s.e.m. Dosing schedule identical to
*Figure 5A*. GEM dosed twice weekly I.P. for the duration of the experiment starting 3 days post-implantation of cancer cells. *=p ≤ 0.05, **=p ≤ 0.01,
***=p ≤ 0.001, ****=p ≤ 0.0001, unpaired t-test. n = 6 tumors per group.

subcutaneously into C57BL/6 mice and tumor growth monitored in response to therapy (*Figure 7E*). Gemcitabine treatment of Panc47 tumors resulted in a significant growth delay that was not further enhanced by combination with either BI 853520 or anti-OX40. However, the triple combination of Gemcitabine + BI 853520 + anti-OX40 did improve response when compared to Gemcitabine alone, Gemcitabine + BI 853520, or Gemcitabine + OX40, and tumors were also smaller than corresponding measurements for BI 853520 + anti-OX40 (*Figure 7D*). Therefore, Gemcitabine exhibits anti-tumor efficacy that can add to that of BI 853520 + anti-OX40.

## Discussion

We show that treatment with a FAK inhibitor can induce potent anti-tumor immune responses in murine models of cancer when used in combination with either endogenous or exogenous signals that promote activation of T-cell co-stimulatory pathways. In particular, we identify CD80, 4-1BB, and OX40 as promising candidate pathways for co-targeting with a FAK inhibitor and show that these combinations can unlock anti-tumor immune responses capable of driving complete regression of at least some mouse tumor models. We provide novel mechanistic insight into the complex immune modulation that occurs in response to a FAK inhibitor, identifying not only changes in immune cell recruitment to tumors, but also in the expression of molecular pathways including PD-L2 and ICOS that fine tune the efficacy of the anti-tumor T-cell response. These data identify new strategies for the development of FAK inhibitors and enhance our understanding of how FAK regulates the immuno-suppressive tumor environment that can modulate response to immunotherapy.

A number of Phase-I clinical trials have now been completed testing FAK inhibitors in patients with a broad range of solid tumor types (*Sulzmaier et al., 2014*; *de Jonge et al., 2019*; *Doi et al., 2019*; *Lee et al., 2015*; *Soria et al., 2012*). Alongside this, there have been a number of studies aimed at defining potential strategies for identification of patients most likely to benefit from a FAK inhibitor. Both Merlin and E-cadherin have been proposed as potential biomarkers in this context (*Hirt et al., 2018*; *Kato et al., 2017*; *Shah et al., 2014*; *Shapiro et al., 2014*). However, a Phase-II clinical trial in which patients with Mesothelioma were treated with the FAK inhibitor Defactinib and tested for Merlin expression was terminated early due to lack of improved efficacy as a maintenance therapy for Merlin-low tumors (ClinicalTrials.gov NCT01870609). To date, the relationship between E-cadherin status and sensitivity to FAK kinase inhibition has not been reported in the clinic. Our data provide a mechanistic justification supporting an alternative strategy for patient stratification based on cancer cell CD80 expression. CD80 has been linked to promoting anti-tumor immune responses in multiple mouse models of cancer (*Marchiori et al., 2019*; *Scarpa et al., 2015*; *Baskar et al., 1993*; *Chen et al., 1992*; *Ganesan et al., 2007*; *Haile et al., 2011*; *Hodge et al., 1994*; *Liu et al., 2001*; *Townsend and Allison, 1993*), and exogenous administration of recombinant CD80 protein can promote anti-tumor immunity (*Horn et al., 2018*). However, CD80 is a ligand for two receptors with opposing functions in regulating T-cell responses, namely CTLA-4 and CD28 (*Linsley et al., 1991*; *Linsley et al., 1990*). CTLA-4, which functions to suppress T-cell responses, has a higher avidity for CD80 (*van der Merwe et al., 1997*) and depletes CD80 availability by the process of trans-endocytosis (*Qureshi et al., 2011*), thereby impacting on CD80 - CD28 interaction and T-cell activation. We show that a FAK inhibitor can shift this balance in favor of CD80 - CD28 interaction, promoting an anti-tumor immune response that is dependent on both CD80 and CD28. The clinical translation of these findings will require an improved understanding of cancer cell CD80 expression in human malignancies. CD80 expressing cancer cells have been identified in human skin SCC (*Miao et al., 2019*) and pre-neoplastic colon lesions (*Marchiori et al., 2019*; *Scarpa et al., 2015*), and our analysis of human cancer cell line transcriptomics data further suggests that cancer cell *CD80* expression is present in a broad range of solid epithelial cancers. However, further investigation will be required in order to fully understand the extent of CD80 protein expression in human malignancies. In addition to solid cancers, we also identified a number of hematological

malignancies that co-express high levels of *CD80* and *PTK2*, highlighting potential opportunities for the development of FAK inhibitors for the treatment of some blood cancers. To our knowledge, FAK kinase inhibitors have never been clinically tested in these tumor types.

Following the success of immune checkpoint inhibitors there has been growing interest in targeting T-cell co-stimulatory pathways as potential cancer therapies. Agonistic antibodies and/or RNA aptamers targeting receptors including 4-1BB, OX40, CD40, GITR, ICOS and CD28 are in pre-clinical development, and have shown anti-tumor activity in mouse models of cancer (*Sanmamed et al., 2015*). Humanized antibodies for several of these receptors have now progressed to clinical testing, with some results suggesting limited anti-tumor efficacy as a monotherapy. For example, a Phase-I clinical trial of the anti-4-1BB antibody PF-05082566 in patients with advanced solid tumors reported an objective response rate of 3.8% (*Segal et al., 2018*), while MOXR0916 (anti-OX40) achieved stable disease in some patients (*Aaron Hansen et al., 2016*). These and other antibodies are now being tested in combination with various other immune and non-immune targeted therapies (*Sanmamed et al., 2015*; *Linch et al., 2015*). Our data suggest that a FAK inhibitor represents a promising candidate for combination with either anti-OX40 or anti-4-1BB antibodies. We propose that it is the complex reprogramming of multiple immune regulatory mechanisms in response to FAK inhibition that complements the immune stimulatory effects of these antibodies in order to fine tune the efficacy of the anti-tumor response. In support of this, we found a novel role for FAK in regulating expression of the immune checkpoint ligand PD-L2 and showed that this may contribute to the improved efficacy of the FAK/anti-OX40 combination. Concurrent administration of anti-PD-1 has previously been reported to suppress the therapeutic effect of anti-OX40 (*Messenheimer et al., 2017*; *Shrimali et al., 2017*). Our findings suggest that these previous observations are likely due to blockade of PD-L1 - PD-1 signaling which is not regulated by FAK. We also identified a novel role for FAK in regulating the expression of ICOS, primarily on CD8$^{eff}$ T-cells, and showed that blocking ICOS ligand could impair the anti-tumor efficacy of both FAK/anti-OX40 and FAK/anti-4-1BB combinations. The ICOS pathway has previously been implicated in regulating the therapeutic response to anti-OX40, but not 4-1BB, leading to the conclusion that ICOS signaling on CD4 T-cells was important for anti-OX40 activity (*Metzger et al., 2016*). Our findings suggest that at least in some circumstances ICOS may also play an important role in regulating CD8 T-cell responses. These disparities may represent differences in the cancer models used, or may perhaps be explained by our observation that anti-OX40 alone only results in a relatively small upregulation of ICOS on CD8$^{eff}$ T-cells, while anti-OX40 in combination with a FAK inhibitor results in a much greater upregulation of ICOS expression. While much of the research into ICOS function has focused on its role in CD4 T-cells, there is evidence supporting a role for ICOS in CD8 T-cell effector function. Ectopic expression of ICOS ligand can result in CD8 T-cell co-stimulation and tumor regression in the absence of CD4 T-cells (*Wallin et al., 2001*). ICOS-deficient patients have a reduced number of CD8 memory T-cells and impaired CD8 T-cell interferon-γ production (*Takahashi et al., 2009*). Therefore, FAK-dependent induction of ICOS expression on CD8$^{eff}$ T-cells may represent an important and novel mechanism contributing to the efficacy of FAK/immunotherapy combinations that warrants further investigation.

Here, we have assessed individual co-stimulatory antibodies in combination with a FAK inhibitor. However, simultaneous administration of multiple co-stimulatory antibodies in combination with a FAK inhibitor may further improve efficacy. Pre-clinical studies have shown that anti-4-1BB when used in combination with anti-OX40 is more effective at boosting CD8 T-cell expansion, effector function, and anti-tumor immunity (*Adler and Vella, 2013*; *Lee et al., 2004*; *Morales-Kastresana et al., 2013*). Indeed, this combination is reported to have synergistic effects that not only enhance CD8 T-cell clonal expansion, but also endow CD8 T-cells with supereffector function (*Lee et al., 2007*). Our data strongly support further investigation of such combinations in conjunction with a FAK inhibitor with a view to clinical translation.

## Materials and methods

### Materials

BI 853520 was provided by Boehringer Ingelheim GmbH. pcDNA3 construct encoding the ORF for murine CD80 was synthesized by GeneArt (Invitrogen). All flow cytometry antibodies used are listed in *Supplementary files 1*, *2* and *3*.

### Cell lines

A selection of murine tumor derived cell lines were used in this study, namely, Squamous Cell Carcinoma cell lines (SCC7.1 and SCC6.2), an MMTV-PyMT mammary tumor cell line (Met01), and *LSL-Kras$^{G12D/+}$;LSL-Trp53$^{R172H/+}$;Pdx-1-Cre* derived pancreatic cancer cell lines (Panc43, Panc47, Panc117). Cells were pathogen tested in September 2016 using the ImpactIII test (Idex Bioresearch) and were negative for all pathogens. Cell lines are routinely tested for mycoplasma every 2–3 months in-house and have never been found to be mycoplasma positive. Cell lines are cultured for no more than 3 months following freeze thawing. SCC cell lines were generated and cultured as previously described (*Serrels et al., 2015*; *Serrels et al., 2012*). SCC6.2 cells stably expressing pcDNA3-CD80 were generated by transfection using Lipofectamine 2000 (Invitrogen) and selection with 0.6 mg/ml geneticin. Met01 and Panc cell lines were cultured in DMEM supplemented with 4500 mg/L glucose, L-glutamine, sodium pyruvate, sodium bicarbonate, and 10% FBS.

### Subcutaneous tumor growth

All animal work were carried out in compliance with UK Home Office guidelines. $1 \times 10^6$ (SCC FAK-wt, SCC FAK-/-, and Met01) or $5 \times 10^5$ (SCC7.1, SCC6.2, Panc43, Panc47, and Panc117) cells were injected subcutaneously into each flank of either FVB/N (SCC and Met01 cell lines) or C57BL/6 mice (Panc cell lines), and tumor growth measured twice weekly using calipers. Animals were euthanized when tumors reached maximum allowed size, or more commonly when signs of ulceration, bleeding, or exudation were evident. For studies involving treatment with BI 853520, drug was prepared in 0.5% carboxymethyl cellulose (Vehicle) (Sigma-Aldrich), and mice were treated at 50 mg/kg QD by oral gavage daily, starting on the day of tumor cell implantation. Isotype control, anti-GITR (clone DTA-1), anti-CD40 (clone FGK4.5), anti-4-1BB (clone LOB12.3), anti-OX40 (clone OX-86), anti-CD28 (clone 37.51), anti-ICOSL (clone HK5.3), anti-CD80 (clone 16-10A1), and anti-PD-L2 (clone TY25) antibodies were dosed twice weekly by intraperitoneal injection at a concentration of 100 µg/mouse diluted in PBS (BioXcell). Animals were visually monitored for signs of toxicity and weighed prior to each dose of BI 853520 or antibody. No signs of toxicity or weight loss were observed. Group sizes ranged from 3 to 5 mice, each bearing two tumors, and tumor volume was calculated in Excel (Microsoft) using the formula V = ½(length x width$^2$). Statistics and graphs were calculated using Prism (GraphPad).

### Tumor growth following Re-Challenge

SCC6.2 cells were injected into both flanks of FVB/N mice and treatment administered as above. Following tumor regression, mice were maintained without treatment for 2 months prior to rechallenge with $5 \times 10^5$ SCC6.2 cells per flank. Tumor growth was measured twice-weekly as described above. At the time of rechallenge, an age-matched control cohort of mice that had not previously been challenged with tumor cells were injected on both flanks using the same cell preparation and tumor growth monitored as above. Tumor volume was calculated as described above.

### CD8$^+$ T cell depletion

Anti-mouse CD8 depleting antibody (clone 53–6.7) and isotype control were purchased from BioXcell. As described previously (*Serrels et al., 2015*), mice were treated with 150 µg of antibody administered by intraperitoneal injection for three consecutive days, followed by a rest period of 3 days. Following this, SCC or Met01 cells were injected subcutaneously into both flanks and T-cell depletion maintained by further administration of 150 µg depleting antibody at 3 day intervals for the remainder of the experiment. Tumor growth was measured twice-weekly as described above.

## FACS analysis

Tumors established following subcutaneous injection of cells into mice were removed at day 12 into DMEM (Sigma-Aldrich). Tumor tissue was mashed using a scalpel and re-suspended in DMEM (Sigma-Aldrich) supplemented with 2 mg/ml collagenase D (Roche) and 40 units/ml DNase1 (Roche). Samples were incubated for 30 min at 37°C, 5% $CO_2$ on an orbital shaker set at 120 rpm, and then pelleted by centrifugation at 1300 rpm for 5 min at 4°C. Samples were re-suspended in 5 ml of red blood cell lysis buffer (Pharm Lysis Buffer, Becton Dickinson) for 10 min at 37°C, pelleted by centrifugation at 1300 rpm for 5 min at 4°C, re-suspended in PBS and mashed through a 70 µm cell strainer using the plunger from a 5 ml syringe. The cell strainer was further washed with PBS. The resulting single cell suspension was pelleted by centrifugation at 1300 rpm for 5 min at 4°C and re-suspended in PBS. This step was repeated twice. The resulting cell pellet was re-suspended in PBS containing Zombie NIR viability dye [1:1000 dilution (BioLegend)] and incubated at 4°C for 30 min then pelleted by centrifugation at 1300 rpm for 5 min at 4°C. Cells were resuspended in FACS buffer (PBS + 1% FBS + 0.1% sodium azide) and pelleted by centrifugation at 1300 rpm for 5 min at 4°C. This step was repeated twice. Cell pellets were resuspended in 100 µl of Fc block [1:200 dilution of Fc antibody (eBioscience) in FACS buffer] and incubated for 15 min. 100 µl of antibody mixture [diluted in FACS buffer (antibody details listed in *supplementary files 1*, *2* and *3*)] was added to each well and the samples incubated for 30 min in the dark. The cells were then pelleted by centrifugation at 1300 rpm for 5 min at 4°C and washed twice with FACS buffer as above. Finally, cells were re-suspended in FACS buffer and analyzed using a BD Fortessa. Data analysis was performed using FlowJo software. Statistics and graphs were calculated using Prism (GraphPad). For flow cytometry analysis of cell lines, growth medium was removed and cells washed twice in PBS. Adhered cells were dissociated from tissue culture flasks by incubating them in enzyme free cell dissociation solution (Millipore) for 10 min at 37°C, 5% $CO_2$, and then scraping with a cell scraper. Cells were pelleted by centrifugation at 1300 rpm for 5 min at 4°C and washed with PBS. This step was repeated twice. Cells were then resuspended in viability dye and stained as above.

## Nanostring analyses

RNA extracts were obtained using a RNeasy kit (Qiagen), following manufacturer's instructions. 100 ng of RNA was analyzed using a mouse nanostring PanCancer Immune Profiling panel as per the manufacturer's instructions. Hybridization was performed for 18 hr at 65°C and samples processed using the nanostring prep station set on high sensitivity. Images were analyzed at maximum (555 fields of view). Data were normalized using nSolver 4.0 software.

## Analysis of CD80 expression in human cancer cell line data

The expression of CD80 and FAK were assessed across the panels of cell lines from the Cancer Cell Line Encyclopedia (*Barretina et al., 2012*) using data downloaded from The Broad Institute portal (https://portals.broadinstitute.org/ccle).

## Generation of bone marrow-derived macrophages (BMDMs)

Bilateral tibias and femurs dissected from FVB/N mice were flushed with 5 ml of DMEM medium supplemented with 10% FBS and 1% Penicillin/Streptomycin into a 50 ml tube, washed in medium once and filtered through a 70 µm cell strainer. Cells were seeded at $1 \times 10^6$ per well in a 6-well plate and cultured in 2 ml of DMEM with 10% FBS and 25 ng/ml recombinant mouse M-CSF for 7 days. BMDMs generated this way were then washed with PBS followed by replacement with fresh media containing recombinant mouse IL-4 (10 ng/ml) and/or BI 853520 (100nM). BMDMs were cultured for a further 48 hr, washed with PBS and harvested using non-enzymatic dissociation buffer, stained with fluorescent conjugated antibodies and analyzed by Flow cytometry as described above.

## Acknowledgements

We would like to thank the Flow Cytometry facility at the University of Edinburgh Centre for Inflammation Research for their help with FACS analysis, the Central Biological Services animal technicians at Little France 2, University of Edinburgh for their assistance with animal research studies, and the

Host and Tumour Profiling Unit at the University of Edinburgh CRUK Centre for help with nanostring analysis.

## Additional information

### Competing interests

Irene C Waizenegger: Employee of Boehringer Ingelheim. The other authors declare that no competing interests exist.

### Funding

| Funder | Grant reference number | Author |
|---|---|---|
| Cancer Research UK | C54352/A22011 | Alan Serrels |
| Boehringer Ingelheim | | Alan Serrels |
| Cancer Research UK | C39669/A25919 | Alan Serrels |

Cancer Research UK had no role in study design, data collection and interpretation, or the decision to submit the work for publication. Boehringer Ingelheim contributed to the study conceptualisation and decision to submit the work for publication.

### Author contributions

Marta Canel, Conceptualization, Formal analysis, Investigation, Methodology, Project administration, Writing - review and editing; David Taggart, Andrew H Sims, David W Lonergan, Formal analysis, Investigation, Methodology; Irene C Waizenegger, Conceptualization, Resources; Alan Serrels, Conceptualization, Formal analysis, Funding acquisition, Investigation, Methodology, Writing - original draft, Project administration, Writing - review and editing

### Author ORCIDs

Marta Canel (iD) http://orcid.org/0000-0002-3390-0705
David Taggart (iD) https://orcid.org/0000-0001-8781-4936
Andrew H Sims (iD) https://orcid.org/0000-0001-9082-3665
David W Lonergan (iD) https://orcid.org/0000-0003-3390-2636
Alan Serrels (iD) https://orcid.org/0000-0003-4992-6077

### Ethics

Animal experimentation: All animal work was carried out in compliance with UK Home Office guidelines under project license number PPL 7008897, and in accordance with the principles of the 3Rs. Every effort was made to minimise suffering.

### Decision letter and Author response

Decision letter https://doi.org/10.7554/eLife.48092.sa1
Author response https://doi.org/10.7554/eLife.48092.sa2

## Additional files

### Supplementary files

- Supplementary file 1. T-cell flow cytometry panel 1.
- Supplementary file 2. T-cell flow cytometry panel 2.
- Supplementary file 3. Non-T-cell flow cytometry panel.
- Supplementary file 4. Markers used to identify tumor infiltrating cell populations.
- Transparent reporting form

## Data availability

All data generated or analysed during this study are included in the manuscript and supporting files. Source data files have been provided for Figure 3.

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
