## [Decision Letter]

**Acceptance summary:**

The major concerns on this study were the clarification of new mechanistic insight particularly on the role of CD80 expression on cancer cells in the FAK inhibitor sensitivity, and the mechanisms of actions for the combination therapy with FAK inhibitors and agonistic antibodies to OX40 or 4-1BB.

In this revised manuscript, as suggested by the reviewers, the authors performed additional in vivo experiments. The results showed that anti-CD80 and anti-CD28 blocking antibody abrogated the anti-tumor activity of FAK inhibitor, indicating the importance of CD80-CD28 interaction in the anti-tumor effects of FAK inhibitor. In addition, CD80 expression in human cancer cells was clarified.

The FAK inhibitor's effect on PL-L2 expression on cancer cells and macrophages was a potential new finding. As suggested by the reviewers, the authors performed additional experiments. FAK inhibitor showed only a minor decrease of PD-L2 on macrophages, indicating that additional in vivo mechanisms exist. IFN-g did not induce PD-L2 on cancer cells, so the effects of FAK inhibitor on PD-L2 in cancer cells was not evaluated.

During these additional experiments, the authors found an additional new mechanism that is ICOS expression on CD8^+^ T-cells and non-Treg CD4^+^ T-cells were markedly increased following FAK inhibitor administration. ICOSL blocking antibody abrogated the anti-tumor effects of combination therapy with FAK inhibitor and OX40 or 4-1BB, indicating involvement of ICOS+ T-cells in this combination therapy.

The effects on the tumor infiltration of CD8^+^ T-cells by FAK inhibitor were found to be dependent on tumor cells, although the authors did not evaluate changes of tumor antigen specific T-cells.

Overall, in the revised manuscript, with new results of additional experiments, the authors clarified the role of CD80 on CD80 positive cancer cells in the sensitivity to FAK inhibitor, and also showed new mechanistic insight of the combination therapy (FAK inhibitor plus agonistic OX40/4-1BB antibodies), including the regulation of PD-L2 and ICOS by FAK inhibitor.

**Decision letter after peer review:**

Thank you for submitting your article "T-cell co-stimulation in combination with targeting FAK drives enhanced anti-tumor immunity" for consideration by *eLife*. Your article has been reviewed by three peer reviewers, and the evaluation has been overseen by a Reviewing Editor and Jeffrey Settleman as the Senior Editor.

The reviewers have discussed the reviews with one another and the Reviewing Editor has drafted this decision to help you prepare a revised submission.

Summary:

Canel et al. present a study where they propose that targeting tumor FAK drives anti-tumor immunity in combination with anti-CD80, anti-OX40, and anti-41BB in models. The authors showed that CD80 expression in cancer cells was correlated with sensitivity to FAK inhibitors, and combination of FAK inhibitors and anti-OX40 or 4-1BB agonistic Ab is also effective for CD80 negative epithelial cancers partly through down-regulation of PD-L2 expression on cancer cells and macrophages.

Essential revisions:

The major role of FAK and FAK inhibitors have been published in the context of cancer immunology and immunotherapy (Jiang et al., 2016; Serrels et al., 2015). The novelty and interest of the present study depends on whether the authors can convincingly show the mechanisms of combinatorial therapies and translational potential. In this regard, this manuscript is still descriptive with excessive association data, and lack of mechanistic analysis. If they had data demonstrating distinct mechanisms by which FAK inhibitor therapy can be utilized along with anti-CD80, anti-OX40, and anti-41BB, this would be an interesting observation, but such data was not provided. Therefore, further investigations are needed as described below to clarify the underlying mechanisms for both CD80 positive and negative cancers in murine and human cancers. Especially further clarification of the role of CD80 on CD80 positive cancer cells and PD-L2 on CD80 negative cancer cells in the sensitivity of FAK inhibitors by using gene manipulation or antibody study are essential for the revision.

The authors showed that CD80 expression in cancer cells was correlated with response to FAK inhibitors by comparing CD80 positive and negative murine cancer cell lines, and the anti-tumor activity was CD8^+^ T-cell dependent. The analysis of human cancer showed that high CD80 expression was only detected in hematological malignancies, not in epithelial cancers. The expression of both CD80 and PTK2 (FAK gene) was shown only in some of lymphomas including Burkitt's and Hodgkin's lymphoma, although the gene expression of CD80 and PTK2 is inversely correlated in general. Therefore, CD80 expression may be a marker for sensitivity to FAK inhibitors, although FAK inhibitors have not been evaluated before for hematologic malignancies. The authors did not evaluate the molecular mechanisms for high sensitivities of CD80+ cancer cells. The authors discussed the possibility that reduction of CTLA-4^+^ Treg by FAK inhibitors may increase the availability of CD80 to CD28 on T-cells, but anti-CD28 agonistic Ab did not enhance anti-tumor activity of FAK inhibitors, so the role of CD80/CD28 interaction in the FAK inhibitor activity is not clear. In human cancers, CD80 is expressed only in some of the lymphomas, not in epithelial cancers. The authors did not use CD80 positive lymphoma models to confirm their sensitivity to FAK inhibitors. So they have insufficient human data indicating how to select and target patient populations, and their suggested translation is not well justified.

For the major conclusion "CD80 expression on tumor cells plays a crucial role in the anti-tumor effects of FAK inhibitor", further studies may be necessary as follows: 1) Could the anti-tumor effects of FAK inhibitor be abrogated by a defect of CD80 on tumor cells by gene editing or blockade of CD80 by the injection with antagonistic mAb? 2) Is the interaction of CD80 with CD28 on T-cells important for the effects of FAK inhibitor? Do the anti-tumor effects of FAK inhibitor disappear in CD28-deficient mice? 3) The authors should investigate the effects of antagonistic anti-CD28 antibody on the tumor regression induced by administration of BI 853520 in various CD80-positive tumor cells including SCC7.1, Met01 or SCC6.2 pcDNA3-CD80, to evaluate a possible involvement of CD28 in BI 853520-mediated reinvigoration of anti-tumor immunity. The same is true for the study of combinatorial additive anti-tumor effects of BI 853520 and agonistic anti-OX40 or anti-4-1BB antibodies on CD80-negative and FAK (PTK2)-high tumor cells.

The authors should investigate various phenotypic changes by the treatment with BI 853520 in tumor micro environment (TME). If the authors would show the data obtained from any in vitro assay that can elucidate effects of BI 853520 on CD80+ FAK (PTK2) high cancer cells, immune cells and many other cells in TME, the significance of this study will be strengthened.

The authors evaluated immune-modulating and anti-tumor activities of FAK inhibitors also for FAK negative epithelial cancers, and found that OX40 or 4-1 BB agonistic Ab synergized with the FAK inhibitor for CD80 negative murine epithelial cancers. However, the authors did not show new mechanism for this combination other than their previously published findings such as reduction of Treg and MDSC/TAM. In this paper, they showed an interesting finding that FAK inhibitor decreased PD-L2, not PD-L1 on cancer cells and MDSC/TAM (Figure 6G and 6I), but the specific role of PD-L2 decrease in the anti-tumor effects by the combination treatment was not investigated. To address this question, anti-tumor effects of FAK inhibitor may be examined in the experiments using PD-L2-KO tumor cells or PD-L2-KO mice. To examine the direct effects of FAK inhibitor on tumor cells to down-regulate PD-L2 expression, in vitro experiment to test PD-L2 expression of tumor cells in the presence of FAK inhibitor needs to be performed. Similar experiments should be performed by using various human cancer cell lines to examine its applicability in clinical setting.

Data in Figure 1H and 1I suggested that the anti-tumor effects of FAK inhibitor are dependent on CD8 T-cells. On the other hand, Figure 6C indicated that treatment with FAK inhibitor did not increase the number of CD8 T-cells. To reconcile these results, the change of CD8 T-cell number in SCC7.1 or Met01 models should be shown. In addition, it may be better to analyze "tumor-specific" rather than "whole" CD8 T-cell number in these experiments.

In the first paragraph of the subsection “Spectrum of responses to BI 853520”, the authors claimed that BI 853520 did not show any off target effects on SCC FAK-wt tumor, because no significant effect on tumor growth was observed in SCC FAK-/- tumor. However because SCC FAK-/- did not grow well in vivo because they are sensitive to endogenous anti-tumor immunity, authors cannot exclude off target effects of BI 853520 by checking effects only on tumor growth.

The authors showed only similarity of cancer stem cell-associated gene expression in SCC7.1 cells, but they did not evaluate stem cell like biological characteristics of SCC7.1 on sensitivity to BI 853520 and subsequent enhancement of tumor immunity. To emphasize the cancer stem cell-like characteristics of SCC7.1, the authors should investigate the high transplantation rate of SCC7.1 in tumor transplantation experiment in vivo.

In Discussion section, the authors often repeated the results and did not describe much about the mechanisms of activated tumor immunity induced by BI 853520 alone or in combination with agonistic anti-OX40 or anti-4-1BB antibodies. If the authors describe this by referring appropriate publications instead of repeating the results, the readers may understand the significance of this study.

---

## [Author Response]

Essential revisions:The major role of FAK and FAK inhibitors have been published in the context of cancer immunology and immunotherapy (Jiang et al., 2016; Serrels et al., 2015). The novelty and interest of the present study depends on whether the authors can convincingly show the mechanisms of combinatorial therapies and translational potential. In this regard, this manuscript is still descriptive with excessive association data, and lack of mechanistic analysis. If they had data demonstrating distinct mechanisms by which FAK inhibitor therapy can be utilized along with anti-CD80, anti-OX40, and anti-41BB, this would be an interesting observation, but such data was not provided. Therefore, further investigations are needed as described below to clarify the underlying mechanisms for both CD80 positive and negative cancers in murine and human cancers. Especially further clarification of the role of CD80 on CD80 positive cancer cells and PD-L2 on CD80 negative cancer cells in the sensitivity of FAK inhibitors by using gene manipulation or antibody study are essential for the revision.

To-date there has been two manuscripts published that detail a role for FAK in regulating recruitment of immune cells with suppressive function into tumors. While this is clearly an important function, we believe the data presented here implies that FAK regulates the anti-tumor immune response in a more extensive and complex manner that has yet to be fully understood. FAK kinase inhibitors are currently in Phase I/II clinical trials, and the major interest here is in their potential as immune oncology agents in combination with other immune targeted therapies. In this context, we identify potential combinations with immune costimulatory agonists that are readily clinically translatable, and provide mechanistic insights into why they are more effective than either monotherapy alone. These findings not only have the potential to impact on the clinical development of FAK inhibitors, but also on immune co-stimulatory agents such as anti-OX40 and anti-4-1BB antibodies which have not shown significant anti-tumour activity as a monotherapy but remain of significant interest to the immune oncology field. We thank the reviewers for their insightful comments that have helped to strengthen the conclusions of this manuscript.

The authors showed that CD80 expression in cancer cells was correlated with response to FAK inhibitors by comparing CD80 positive and negative murine cancer cell lines, and the anti-tumor activity was CD8^+^ T-cell dependent. The analysis of human cancer showed that high CD80 expression was only detected in hematological malignancies, not in epithelial cancers. The expression of both CD80 and PTK2 (FAK gene) was shown only in some of lymphomas including Burkitt's and Hodgkin's lymphoma, although the gene expression of CD80 and PTK2 is inversely correlated in general. Therefore, CD80 expression may be a marker for sensitivity to FAK inhibitors, although FAK inhibitors have not been evaluated before for hematologic malignancies. The authors did not evaluate the molecular mechanisms for high sensitivities of CD80+ cancer cells. The authors discussed the possibility that reduction of CTLA-4^+^ Treg by FAK inhibitors may increase the availability of CD80 to CD28 on T-cells, but anti-CD28 agonistic Ab did not enhance anti-tumor activity of FAK inhibitors, so the role of CD80/CD28 interaction in the FAK inhibitor activity is not clear. In human cancers, CD80 is expressed only in some of the lymphomas, not in epithelial cancers. The authors did not use CD80 positive lymphoma models to confirm their sensitivity to FAK inhibitors. So they have insufficient human data indicating how to select and target patient populations, and their suggested translation is not well justified.

The analysis of human cancer cell line data (Figure 4) does not show that *CD80* is only detected in hematological malignancies, rather it shows that *CD80* is broadly expressed in human cancer cell lines but is generally higher in hematological malignancies. This interpretation is perhaps our fault for putting too much emphasis on hematological malignancies and the manuscript text has been modified as detailed in response to comments below to reflect the broad expression profile identified. The role of CD80 and potential mechanisms are also addressed below.

For the major conclusion "CD80 expression on tumor cells plays a crucial role in the anti-tumor effects of FAK inhibitor", further studies may be necessary as follows: 1) Could the anti-tumor effects of FAK inhibitor be abrogated by a defect of CD80 on tumor cells by gene editing or blockade of CD80 by the injection with antagonistic mAb? 2) Is the interaction of CD80 with CD28 on T-cells important for the effects of FAK inhibitor? Do the anti-tumor effects of FAK inhibitor disappear in CD28-deficient mice? 3) The authors should investigate the effects of antagonistic anti-CD28 antibody on the tumor regression induced by administration of BI 853520 in various CD80-positive tumor cells including SCC7.1, Met01 or SCC6.2 pcDNA3-CD80, to evaluate a possible involvement of CD28 in BI 853520-mediated reinvigoration of anti-tumor immunity. The same is true for the study of combinatorial additive anti-tumor effects of BI 853520 and agonistic anti-OX40 or anti-4-1BB antibodies on CD80-negative and FAK (PTK2)-high tumor cells.

To further investigate the importance of CD80 in the anti-tumor effects of the FAK inhibitor we have added two in vivo experiments (Figure 2E) using an antagonistic anti-CD80 antibody and an antagonistic anti-CD28 antibody. These data clearly show that blocking either CD80 or CD28 can abrogate the effects of the FAK inhibitor. It was not feasible to undertake these studies in multiple tumor models and so we have focused on the SCC7.1 tumor model which undergoes complete CD8 T-cell dependent regression in response to a FAK inhibitor. Furthermore, we have added flow cytometry data showing that treatment of SCC7.1 tumors with the FAK inhibitor results in a decrease in CTLA-4 expressing immune cells and an increase in CD80 surface expression on non-immune cells (Figure 2C, D). These findings further support the proposed mechanism i.e. FAK inhibition reshapes the tumor microenvironment to favor CD80: CD28 interaction through depleting immune cells such as regulatory T-cells that express CTLA-4.

*The authors should investigate various phenotypic changes by the treatment with BI 853520 in tumor micro environment (TME). If the authors would show the data obtained from any* in vitro *assay that can elucidate effects of BI 853520 on CD80+ FAK (PTK2) high cancer cells, immune cells and many other cells in TME, the significance of this study will be strengthened.*

Figure 6, Figure 6—figure supplements 3, 4, 5 and 6 detail a number of changes that occur in the tumor microenvironment in response to BI 853520. Figure 6—figure supplement 7 also shows the effects of BI 853520 on PD-L2 expression by macrophages in vitro. In terms of the effects of BI 853520 on CD80^+^ cancer cells, it is not clear what useful information would be generated from in vitro studies. The new and existing data presented in the manuscript implies that the mechanism of action of BI 853520 on CD80^+^ tumors is likely through shifting the balance of CD80 signaling in favor of CD28 and T-cell activation.

*The authors evaluated immune-modulating and anti-tumor activities of FAK inhibitors also for FAK negative epithelial cancers, and found that OX40 or 4-1 BB agonistic Ab synergized with the FAK inhibitor for CD80 negative murine epithelial cancers. However, the authors did not show new mechanism for this combination other than their previously published findings such as reduction of Treg and MDSC/TAM. In this paper, they showed an interesting finding that FAK inhibitor decreased PD-L2, not PD-L1 on cancer cells and MDSC/TAM (Figure 6G and 6I), but the specific role of PD-L2 decrease in the anti-tumor effects by the combination treatment was not investigated. To address this question, anti-tumor effects of FAK inhibitor may be examined in the experiments using PD-L2-KO tumor cells or PD-L2-KO mice. To examine the direct effects of FAK inhibitor on tumor cells to down-regulate PD-L2 expression,* in vitro *experiment to test PD-L2 expression of tumor cells in the presence of FAK inhibitor needs to be performed. Similar experiments should be performed by using various human cancer cell lines to examine its applicability in clinical setting.*

We have added new data investigating the role of PD-L2 in the combinatorial activity of the FAK inhibitor with either OX40 or 4-1BB.

We have used an anti-PD-L2 blocking antibody to determine the potential contribution of PD-L2 regulation on the activity of anti-OX40 and anti-4-1BB antibodies (Figure 6J, K). These data show individual tumor volumes and suggest that PD-L2 may contribute to the enhanced activity of OX40 with a FAK inhibitor but is less likely to contribute to combination with anti-4-1BB. The effects of anti-PD-L2 on anti-OX40 activity are not statistically significant. However, after 21 days 5 tumors in this group have undergone complete regression while none in the OX40 alone group have completely regressed. We believe that this is indicative of a role for PD-L2 in contributing to the enhanced activity of a FAK inhibitor in combination with anti-OX40.

We have addressed whether a FAK inhibitor regulates PD-L2 expression on macrophages directly (Figure 6—figure supplement 7). These studies have been performed using in vitro bone-marrow derived macrophages +/- treatment with interleukin-4 to stimulate PD-L2 expression. We observed a small decrease in IL-4 stimulated PD-L2 expression on macrophages following treatment with a FAK inhibitor. This regulation is not as extensive as we observe in vivo, implying that additional mechanisms may contribute to FAK-dependent regulation of PD-L2 expression in tumors. Similar studies were also undertaken using a variety of cancer cells, however none of these expressed PD-L2 in response to IL-4 stimulation.

We have also added new data showing a novel role for FAK in regulating ICOS expression, primarily on CD8^eff^ T-cells, and have investigated the contribution of this to the combinatorial activity of the FAK inhibitor with OX40 and 4-1BB. Figure 6D shows that OX40 treatment increases ICOS expression on CD8^eff^ T-cells and that this is markedly enhanced when OX40 is combined with the FAK inhibitor. An increase in ICOS expression is also observed on non-Treg CD4 T-cells (Figure 6—figure supplement 4). Further, regulation of ICOS on CD8^eff^ T-cells in response to the FAK inhibitor is not restricted to combination with OX40, as we also identified this to occur in CD80+ SCC7.1 tumors (Figure 6—figure supplement 5). Using an ICOS ligand blocking antibody we show that ICOS plays an important role in the anti-tumor efficacy of both the FAK inhibitor in combination with OX40 and 4-1BB (Figure 6L, M).

These new data imply that regulation of PD-L2 and ICOS contributes to the mechanisms underlying improved anti-tumor activity of the FAK inhibitor in combination with OX40, and that ICOS plays an important role in the mechanism underpinning the combination with 4-1BB. Further, these studies are the first to identify a role for FAK in regulating molecular pathways, as opposed to immune cell recruitment, that contribute to immune suppression/activation and highlight the complexity of FAK-dependent immune modulation.

Data in Figure 1H and 1I suggested that the anti-tumor effects of FAK inhibitor are dependent on CD8 T-cells. On the other hand, Figure 6C indicated that treatment with FAK inhibitor did not increase the number of CD8 T-cells. To reconcile these results, the change of CD8 T-cell number in SCC7.1 or Met01 models should be shown. In addition, it may be better to analyze "tumor-specific" rather than "whole" CD8 T-cell number in these experiments.

We have previously published (Serrels et al., 2015) that treatment with a FAK inhibitor can increase CD8 T-cell number in the tumor model used in Figure 1A which also expresses CD80. The tumor models used in Figures 1B, C also express CD80 and therefore express a T-cell co-stimulatory signal that will drive T-cell expansion. In Figures 2C, D we show that the CD80, CTLA-4 pathway is modulated by a FAK inhibitor, likely in favor of CD80-CD28 interaction. This pathway is not present in CD80^-^ tumors such as that used in Figure 6. Hence, in the absence of a T-cell co-stimulatory signal we do not see increased CD8 T-cell infiltration into tumors in response to a FAK inhibitor. The additional data included in Figure 6D, Figure 6—figure supplements 4, 5 also imply that a FAK inhibitor may regulate T-cell effector function through increasing expression of T-cell co-stimulatory receptors such as ICOS, and so it is not as simple as just increasing T-cell numbers. For example, Figure 6—figure supplement 5 shows that a FAK inhibitor increases ICOS expression on CD8^eff^ T-cells in the CD80+ SCC7.1 tumors, while it does not in the SCC6.2 tumors which are negative for a T-cell co-stimulatory signal such as CD80 (Figure 6D).

*In the first paragraph of the subsection “Spectrum of responses to BI 853520”, the authors claimed that BI 853520 did not show any off target effects on SCC FAK-wt tumor, because no significant effect on tumor growth was observed in SCC FAK-/- tumor. However because SCC FAK-/- did not grow well* in vivo *because they are sensitive to endogenous anti-tumor immunity, authors cannot exclude off target effects of BI 853520 by checking effects only on tumor growth.*

We have removed this sentence.

*The authors showed only similarity of cancer stem cell-associated gene expression in SCC7.1 cells, but they did not evaluate stem cell like biological characteristics of SCC7.1 on sensitivity to BI 853520 and subsequent enhancement of tumor immunity. To emphasize the cancer stem cell-like characteristics of SCC7.1, the authors should investigate the high transplantation rate of SCC7.1 in tumor transplantation experiment* in vivo.

The purpose of the data presented in Figure 3 was not to prove that the SCC7.1 cells that respond to the FAK inhibitor are stem cells, but rather to show that they exhibit a gene expression profile similar to a population of CD80^+^ skin cancer cells already reported by others to be present in murine and human cancers. We therefore did not undertake limiting dilution assays as we believe this would represent an unnecessary use of animals that on this occasion would not strengthen the key conclusions from this figure. Rather, we have used flow cytometry to confirm the expression of CD34 and Integrin alpha-6, the key markers of the reported skin cancer cell population reported in Miao et al., 2019.

In Discussion section, the authors often repeated the results and did not describe much about the mechanisms of activated tumor immunity induced by BI 853520 alone or in combination with agonistic anti-OX40 or anti-4-1BB antibodies. If the authors describe this by referring appropriate publications instead of repeating the results, the readers may understand the significance of this study.

We apologize for the poor Discussion and have redrafted this to be less repetitive of the results and better discuss the findings in relation to the broader literature.